# Monoculture in Matching Markets

**Kenny Peng**
Cornell Tech
kennypeng@cs.cornell.edu

**Nikhil Garg**
Cornell Tech
ngarg@cornell.edu

## Abstract

Algorithmic monoculture arises when many decision-makers rely on the same algorithm to evaluate applicants. An emerging body of work investigates possible harms of such homogeneity, but has been limited by the challenge of incorporating market effects in which the preferences of many applicants and decision-makers jointly interact to determine outcomes. Addressing this challenge, we introduce a tractable theoretical model of algorithmic monoculture in a two-sided matching market with many participants. We use the model to analyze outcomes under monoculture (when decision-makers all evaluate applicants using a common algorithm) and under polyculture (when decision-makers evaluate applicants independently). All else equal, monoculture (1) selects less-preferred applicants when noise is well-behaved, (2) matches more applicants to their top choice, though individual applicants may be worse off and have higher variance outcomes depending on their value to decision-makers, and (3) is more robust to disparities in the number of applications submitted. Overall, our approach strengthens, challenges, and broadens the scope of the existing monoculture literature.

## 1 Introduction

What happens when many decision-makers—in consequential domains like employment and college admissions—all evaluate applicants using the *same* algorithm? This possibility has been termed "algorithmic monoculture"—a name that references the problems arising in agriculture when a single crop is farmed extensively, amplifying the risk of correlated widespread failures. A recent line of work considers how a lack of variety in algorithms may warrant similar concerns; this concern may be especially important if many decision-makers use the same large language models.

Initial inquiries into the consequences of algorithmic monoculture have identified potential harms to both decision-makers and applicants. In seminal work initiating the formal study of algorithmic monoculture, Kleinberg and Raghavan [35] demonstrated that firms relying on a common algorithm may hire weaker applicants than when using idiosyncratic (but less individually accurate) hiring processes. On the applicant side, an emerging set of work highlights the concern of *systemic exclusion*, which occurs when an applicant is denied from all opportunities [16, 11, 43, 31].

This emerging literature has been limited by the challenge of incorporating *market* effects. Domains where monoculture may be especially impactful—such as employment and college admissions—share some common properties. First, participants on both sides of the market compete: workers compete over a limited number of job openings, and firms compete over workers who can each accept only one offer. Second, outcomes are determined by the preferences of participants on both sides: roughly speaking, a match is formed only if both a worker and firm prefer each other over their other options. Analyzing these interactions across many participants can be theoretically challenging; indeed, the existing monoculture literature in the ML community considers neither markets with many participants nor two-sided interactions.

38th Conference on Neural Information Processing Systems (NeurIPS 2024).

The technical contribution of our paper is a matching markets model that can be used to study algorithmic monoculture. We analyze equilibrium outcomes (i.e., stable matchings) that emerge under *monoculture* (when firms each use the same method to evaluate applicants) and *polyculture* (when firms use idiosyncratic methods). The model builds on the popular continuum model formalized by Azevedo and Leshno [10], where stable matchings are characterized by a cutoff structure in which applicants are matched to their favorite firm among those where their *score* exceeds the firm's *cutoff*. In the model we introduce, an applicant's score is equal to an *estimated value* that is a noisy realization of their *true value*. Under monoculture, all firms use a single shared estimate of an applicant's value; under polyculture, firms obtain separate, independently drawn estimates of the applicant's value. Our theoretical results are shown in a setting with a symmetric setup with a large number of homogeneous firms. This allows for tractability, as well as clear intuition. We demonstrate empirically that our theoretical results hold in markets with heterogeneous firms, as well as when firms use ML models to rank applicants.

We show how modeling market effects with many participants advances the present understanding of monoculture in several ways, with Figure 1 illustrating our first two theoretical results:

- (Theorem 1) First, as long as noise is well-behaved (i.e., with lighter-than-exponential tail), essentially only the most preferred applicants are hired under polyculture when there are many firms. This shows that the existing (relatively modest) effect found by Kleinberg and Raghavan [35] with two firms can be fully strengthened when considering many firms. Our result gives a "wisdom of the crowds"-like result, where even when individual firms are noisy, the market outcome can behave as if firms each had perfect information about applicants—even despite firms not explicitly sharing any information. Polyculture can thus significantly outperform monoculture in terms of firm welfare, even when the idiosyncratic evaluation processes of firms are much noisier than the shared algorithm used under monoculture.

- (Theorem 2) Second, we challenge—and add nuance to—existing work suggesting that monoculture harms applicants. In fact, we show that *on average*, applicants are better off under monoculture than polyculture, in the sense that on average, applicants are matched to firms that they more prefer. On the other hand, the *individual* preference of an applicant for monoculture or polyculture depends on their true value. Some applicants are strictly more likely to be matched to their top choice under monoculture, but strictly less likely to match at all—and so have higher *variance* of outcomes conditional on true value; for these applicants, their preference could depend on other factors such as risk aversion. This finding can also be interpreted as showing that monoculture presents a greater risk of systemic exclusion [16, 11] to more qualified applicants who might be overlooked by a single algorithm. However, it does not pose a greater risk of systemic exclusion *overall*.

- (Theorem 3) Third, we expand the scope of existing analysis by extending our model to incorporate *differential application access*: when applicants submit a varying number of applications. This addresses an equity concern especially salient in college admissions, where the cost of application can be high.[1] It also reveals how different application systems incentivize applicants to submit more applications. We find that applicants who submit many applications benefit much more under polyculture than under monoculture. Consequently, differences in the number of applications submitted harms college and firm welfare more under polyculture than monoculture. Overall, monoculture is more robust than polyculture with respect to differential application access.

Our approach strengthens, challenges, and broadens the scope of the monoculture literature. The effects we study emerge in large markets with two-sided preferences, a standard setting for algorithmic monoculture (e.g., prior works often motivate monoculture through the lens of hiring). Moreover, the cutoff structure inherent in Azevedo and Leshno [10] results in clean intuition for each of our results—as well as precise characterizations at the level of individual applicants. In this way, we bridge the literature on monoculture and matching markets, and we anticipate that the matching literature can provide further insight into the emerging study of monoculture.

The paper proceeds as follows. In Section 2, we consider related work in the algorithmic monoculture and matching markets literature (we give an extended discussion of related work in Appendix B). In

---

[1]Application fees to U.S. colleges were as high as 100 dollars in the 2022-2023 cycle [44]. Moreover, the Common Application reported that the number of applications submitted by students varied significantly in the 2021-2022 cycle; for example, high-volume applicants who submitted 15 or more applications were 2.5x as likely to have attended a private high school [34].

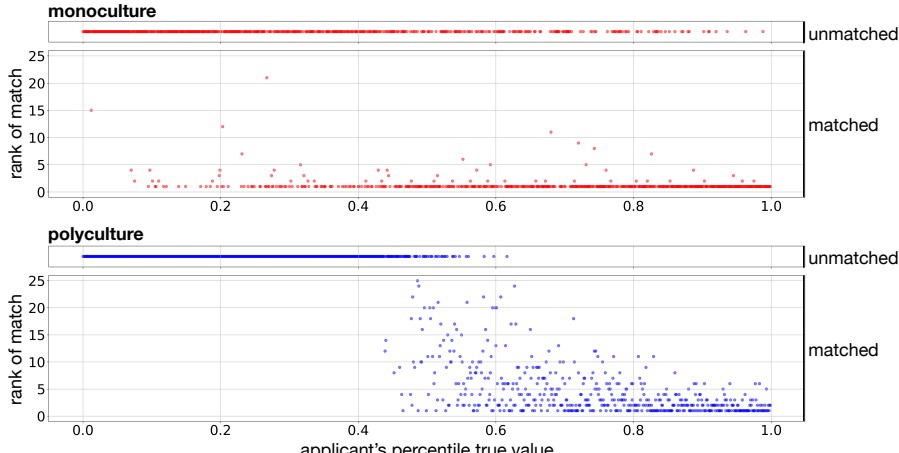

**Figure 1:** Under monoculture, the set of matched applicants is more noisy. However, applicants are more likely to match to their top choices. Roughly speaking, the use of a single algorithm means that applicants who match tend to "do well" across all firms; however, high-value applicants may not match due to a single poor evaluation.

Section 3, we introduce our model and establish some preliminary results. In Section 4, we introduce Theorem 1 and Theorem 2 (we give an extended model and establish Theorem 3 in Appendix A). In Section 5, we provide an overview of our computational experiments; in particular, we test our theoretical predictions in an experiment where firms use ML models (trained on the ACSIncome dataset [19]) to evaluate applicants. Proofs of all theoretical results are given in Appendix E.

## 2   Related Work

**Algorithmic Monoculture in ML**   Our model extends Kleinberg and Raghavan [35]; they analyze the case where two decision-makers each hire one applicant, while we consider many decision-makers who can each hire many applicants. Theorem 1 in the present work shows that the effect found in Kleinberg and Raghavan [35], where polyculture can result in higher total accuracy than monoculture, *is fully strengthened when considering many decision-makers*.

Our two-sided matching markets model also makes it easy to simultaneously study *applicant* welfare, which has recently been of concern. Creel and Hellman [16] argue that *systemic exclusion*, in which applicants are systematically denied all opportunities, can pose significant harm. Bommasani et al. [11] and Toups et al. [43] analyze this effect empirically by examining the level of homogeneity in decision-making across algorithms. One limitation of this existing work is that it does not consider *market-level* effects, which are present in domains such as hiring and college admissions. For example, if the total number of available positions is fixed and if firms intend to fill all of these positions, then the total rate of systemic exclusion does not change (the number of people that do not end up receiving a position is fixed). We show *which* applicants are at a greater risk of systemic exclusion under monoculture. Our model gives a tractable way to theoretically analyze market outcomes. In Section 5, we further show how empirical approaches can be adapted accordingly under our model.

**Matching Markets and School Choice**   By analyzing algorithmic monoculture through a matching markets framework, our results reveal connections between algorithmic monoculture and questions that have been considered in the matching markets (and school choice) literature. For example, Castera et al. [13] analyzes statistical discrimination in stable matching. In their model, the values of applicants from two groups are estimated by two decision-makers, potentially with correlation (monoculture). Like us, they leverage a continuum model of applicants [10]. Their findings focus on aggregate measures of applicant-side welfare across groups. They show, for example, that the group of applicants evaluated with less correlation is matched with higher probability. More directly related to our work, they show that decreased correlation (even in one group), reduces the number of applicants who receive their top choice; this parallels our finding in Theorem 2(ii), which implies that

fewer applicants are matched to their top choice under polyculture. This finding reflects a general result developed in the school choice literature: that increasing correlation in how decision-makers evaluate applicants also increases the total number of applicants matched to their top choice. It is well-established that in school matching, the use of a single lottery number (rather than an independent lottery per school) results in more students being matched to their top choices and is more robust to differences in the number of listed schools [2, 18, 9, 7, 5]. A key distinguishing feature of the model studied here and by [35] is using approximate values to determine which applicants are matched, but to analyze the matching using a different set of true (latent) values.

Interestingly, the matching markets and school choice literature generally shows the *benefits* of increased correlation in how applicant evaluation, while the algorithmic monoculture literature describes primarily *harms*. Our work helps reconcile these competing intuitions. In comparison to the matching markets literature, we assume that each applicant has some ground truth value, and that decision-makers noisily rank students based on this value. Consequently, we can analyze outcomes for *individual* applicants, whereas the markets literature primarily considers homogeneous applicants. This reveals the relevant trade-offs between monoculture and polyculture. For example, we show (Theorem 2) that increased correlation (monoculture) increases the probability that an applicant is matched to their top choice, but can also increase the probability that a high-value applicant does not match at all. Our analysis with many firms and applicants also reveals market *emergent* phenomena not apparent in smaller markets. In follow-up work, we show that Theorem 1 extends to a setting with arbitrary student preferences and college capacities [40]. There, it is shown that when noise is long-tailed, the *opposite* effect can occur, where students match uniformly at random under polyculture. See Appendix B for an extended discussion of related work.

## 3  Model and Preliminaries

Our model builds on Azevedo and Leshno [10], with a continuum of applicants and a finite number of firms (or colleges). An applicant is identified by their true *value*, which is distributed on $\mathbb{R}$ according to a probability measure $\eta$. Let $\mathcal{F} = \{1, 2, \cdots, m\}$ be the set of all firms. Firms have total capacity $S < 1$ (there are fewer positions than applicants) and each firm has the same capacity $\frac{S}{m}$. In our setup, firms have only *estimates* of applicant true values, instead of direct access to the true values. An applicant with value $v$ is associated with an independent random variable $\theta(v)$, characterizing their preferences over firms and firms' estimated values for that applicant. The realization of $\theta(v)$ is their *type*, which lies in the set of applicant types $\Theta := \mathcal{R} \times \mathbb{R}^m$, where $\mathcal{R}$ is the set of preference orderings over the firms. If $\theta(v) = \theta = (\succ^\theta, e^\theta)$, then $\succ^\theta$ is the preference ordering of $v$ over firms and $e_f^\theta$ is the *estimated value* of the applicant at firm $f$. (In this way, any measurable subset of student values induces a measure over $\Theta$.) Firms all prefer applicants with higher true values, but in practice, rank applicants based on the estimated values given by $\theta(v)$.

A *matching* is a function $\mu : \Theta \to \mathcal{F} \cup \{\emptyset\}$ that satisfies the capacity constraints

$$\int_{\mathbb{R}} \Pr[\mu(\theta(v)) = f] \, d\eta(v) = \frac{S}{m}, \qquad \forall f \in \mathcal{F}, \tag{1}$$

as well as two additional technical assumptions: that $\mu^{-1}(f)$ is measurable and that the set $\{\theta : \mu(\theta) \prec^\theta f\}$ is open. Then $\mu(\theta(v))$ is a random variable indicating where an applicant with value $v$ is matched, with $\mu(\theta(v)) = \emptyset$ indicating that the applicant is not matched. When it is clear what $\theta$ is, we will abuse notation to write $\mu(v)$ instead of $\mu(\theta(v))$ to denote the random variable representing where an applicant with value $v$ is matched. A matching is *stable* if there does not exist a pair $\theta \in \Theta, f \in \mathcal{F}$ such that $f \succ^\theta \mu(\theta)$, and $e_f^\theta > e_f^{\theta'}$ for some $\theta' \in \mu^{-1}(f)$. In other words, there is no applicant-firm pair that would jointly prefer to defect from the current matching to be matched with each other.

**A Cutoff Characterization of Stable Matching**  A benefit of the continuum model is that stable matchings are characterized by a tractable cutoff structure [10]. Given a vector of *cutoffs* $P = (P_1, \cdots, P_m)$, an applicant with type $\theta$ can *afford* firm $f$ if $e_f^\theta \geq P_f$. The applicant's *demand* at $P$ is $D^\theta(P)$, the applicant's most preferred firm (according to $\succ^\theta$) among those they can afford. If they cannot afford any firm, $D^\theta(P) = \emptyset$. The *aggregate demand* of a firm $f$ at $P$ is

$$D^f(P) := \int_{\mathbb{R}} \Pr[D^{\theta(v)}(P) = f] \, d\eta(v), \tag{2}$$

the measure of applicants who demand it. A vector of cutoffs $P$ is *market clearing* if and only if $D^f(P) = \frac{S}{m}$ for all $f \in m$. Notably, stable matchings are in one-to-one correspondence with market-clearing cutoffs:

**Lemma 1** (Supply and Demand Lemma [10]). *A matching $\mu$ is stable if and only if there exists a vector of market-clearing cutoffs $P$ such that $\mu(\theta) = D^\theta(P)$.*

The characterization of stable matchings given by the continuum model is summarized as follows. Applicants each have an estimated value at each firm. Firms each have cutoffs. An applicant is then matched to their most preferred firm among those at which an applicant's estimated value is above the cutoff. Next, we show how we can instantiate monoculture and polyculture within the model.

**Instantiating Monoculture and Polyculture** We instantiate our model by specifying $\theta(v)$ for monoculture and polyculture. In what follows, we consider a fixed *noise distribution* $\mathcal{D}$.

- In *monoculture*, we take the random variable

$$\theta_{\mathrm{mono}}(v) := (\succ, (v + X, v + X, \cdots, v + X)), \tag{3}$$

  where $\succ$ is drawn uniformly at random from $\mathcal{R}$ and $X \sim \mathcal{D}$. In other words, applicants have preferences distributed uniformly at random, and each applicant has a single estimated value $v + X$ that is shared across firms. This noisy estimated value equals the applicant's value perturbed by noise drawn from $\mathcal{D}$.

- In *polyculture*, we take the random variable

$$\theta_{\mathrm{poly}}(v) := (\succ, (v + X_1, v + X_2, \cdots, v + X_m)), \tag{4}$$

  where $\succ$ is drawn uniformly at random from $\mathcal{R}$ and $X_1, X_2, \cdots, X_m \sim \mathcal{D}$ are i.i.d.. Again, applicants have preferences distributed uniformly at random, but now each applicant has a distinct, independently drawn estimated value at each firm.[2]

**Preliminary Analysis** We establish some basic results. First, we show that under both monoculture and polyculture, stable matchings are characterized by identical cutoffs at each firm.[3]

**Lemma 2** (Equal Cutoffs Lemma). *If $\theta \in \{\theta_{\mathrm{mono}}, \theta_{\mathrm{poly}}\}$, then there is a unique vector of market-clearing cutoffs $P$, and $P_1 = P_2 = \cdots = P_m$.*

We denote these unique *shared cutoffs* by $P_{\mathrm{mono}}$ and $P_{\mathrm{poly}}$, respectively, with corresponding stable matchings $\mu_{\mathrm{mono}}$ and $\mu_{\mathrm{poly}}$. Under our abuse of notation, $\mu_{\mathrm{mono}}(v) := \mu_{\mathrm{mono}}(\theta_{\mathrm{mono}}(v))$ and $\mu_{\mathrm{poly}}(v) := \mu_{\mathrm{poly}}(\theta_{\mathrm{poly}}(v))$ are random variables representing where $v$ matches. This cutoff structure allows for clean analysis of stable matchings. The next proposition follows directly from the Supply and Demand Lemma (Lemma 1) and the Equal Cutoffs Lemma (Lemma 2).

**Proposition 3.** *For all $v \in \mathbb{R}$,*

*(i) the probability an applicant with value $v$ is matched under monoculture is*

$$\Pr[\mu_{\mathrm{mono}}(v) \in \mathcal{F}] = \Pr[v + X > P_{\mathrm{mono}}], \tag{5}$$

*where $X \sim \mathcal{D}$, and*

*(ii) the probability an applicant with value $v$ is matched under polyculture is*

$$\Pr[\mu_{\mathrm{poly}}(v) \in \mathcal{F}] = \Pr[v + \max_{f \in \mathcal{F}} X_f > P_{\mathrm{poly}}], \tag{6}$$

*where $X_1, \cdots, X_m \overset{\mathrm{iid}}{\sim} \mathcal{D}$.*

---

[2] We make a basic technical assumption on the probability measure $\eta$ and the probability distribution $\mathcal{D}$. Let $\pi$ be the probability measure associated with $\mathcal{D}$; we assume that both $\eta$ and $\pi$ have *connected support*, where we mean that the smallest closed set of measure 1 is an interval. Let these respective intervals be $[V_-, V_+]$ and $[X_-, X_+]$. This assumption ensures a unique stable matching.

[3] The result intuitively follows from the existence of unique market-clearing cutoffs under reasonable conditions (Theorem 1 in [10]). A complete proof is provided in the appendix.

Proposition 3 contrasts the structure of stable matchings in the monoculture and polyculture economies. In monoculture, applicants match depending on a single noisy estimated value. In polyculture, applicants match depending on the *maximum* of $m$ noisy estimated values.

Proposition 3 also reveals that the shared cutoff is lower under monoculture than polyculture. This explains the puzzle of why it is not the case that fewer applicants are matched overall under monoculture: when estimated values at firms become more correlated, the acceptance cutoff decreases:

**Corollary 4** (Lower Cutoff Under Monoculture). *When $m \geq 2$, the shared cutoff in an economy is lower under monoculture than under polyculture: i.e., $P_{\mathrm{mono}} < P_{\mathrm{poly}}$.*

Intuitively, the result holds because under polyculture, applicants have multiple chances at getting an estimated value that is higher than the shared cutoff; so in order for the same number of applicants to be matched under monoculture and polyculture, the cutoff under monoculture must be lower.

**Definition: Maximum-Concentrating Noise**    We now introduce basic definitions that will be useful for reasoning about the noise distribution $\mathcal{D}$. In particular, we define a particular class of noise distributions that we will focus our later analysis on. The **maximum order statistic** of $n$ draws from a distribution $\mathcal{D}$ is defined as the random variable $X^{(n)} := \max\{X_1, X_2, \cdots, X_n\}$ where $X_1, \cdots, X_n \overset{\text{iid}}{\sim} \mathcal{D}$. The maximum order statistic plays an important role in our analysis, since the probability an applicant with value $v$ is matched under polyculture (6) can be rewritten as

$$\Pr[\mu_{\mathrm{poly}}(v) \in \mathcal{F}] = \Pr[v + X^{(m)} > P_{\mathrm{poly}}]. \tag{7}$$

A distribution $\mathcal{D}$ is **maximum concentrating** if for all $\varepsilon > 0$,

$$\lim_{n \to \infty} \Pr\left[|X^{(n)} - \mathbb{E}[X^{(n)}]| > \varepsilon\right] = 0. \tag{8}$$

In other words, a distribution is maximum concentrating if its maximum order statistic satisfies a "weak law of large numbers." These include the uniform and Gaussian distributions, as well as (roughly speaking) distributions with an upper tail that is "lighter than exponential."[4]

# 4    Main Results: Firm and Applicant Welfare

## 4.1    Firm welfare

In this section, we focus on the perspective of firms, who would like to admit the most-preferred applicants (i.e., those with the highest true values). To do this, we analyze the probability that an applicant with a given value is matched. As it turns out, strong behavior emerges under polyculture as the number of firms $m$ in the economy grows large. In particular, when the noise distribution $\mathcal{D}$ is maximum-concentrating, only the highest-value applicants are matched; as a function of an applicant's value, the probability that an applicant is matched approaches a step function when $m$ approaches $\infty$. This, in turn, implies that the matching under polyculture approaches optimal.

Define $v_S$ to be the unique value such that $\eta((v_S, \infty)) = S$, i.e., the threshold at which the mass of students with value above $v_S$ equals the total firm capacity. Note that in the firm-optimal matching, only applicants with value greater than $v_S$ are matched. We may state our first main result.

**Theorem 1** (Wisdom of the Crowds in Polyculture but not Monoculture). *Suppose $\mathcal{D}$ is maximum concentrating. Then:*

*(i) Under polyculture, the probability an applicant is matched approaches $0$ or $1$ as $m \to \infty$:*

$$\lim_{m \to \infty} \Pr[\mu_{\mathrm{poly}}(v) \in \mathcal{F}] = \begin{cases} 0 & v < v_S \\ 1 & v > v_S \end{cases}, \tag{9}$$

*and firm welfare approaches optimal.*

*(ii) Under monoculture, the probability an applicant is matched, $\Pr[\mu_{\mathrm{mono}}(v) \in \mathcal{F}]$, is constant in $m$, and firm welfare is constant and suboptimal.*

---

[4]In general, it is possible to determine if a distribution is maximum concentrating using the Fisher-Tippett-Gnedenko theorem, a cornerstone of Extreme Value Theory (see [24] for a detailed exposition).

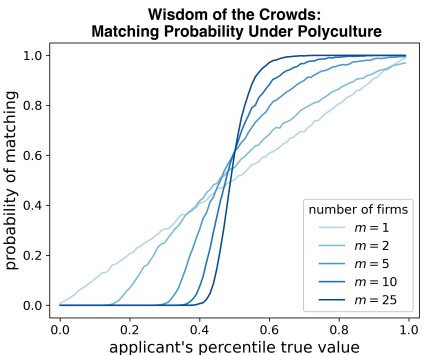
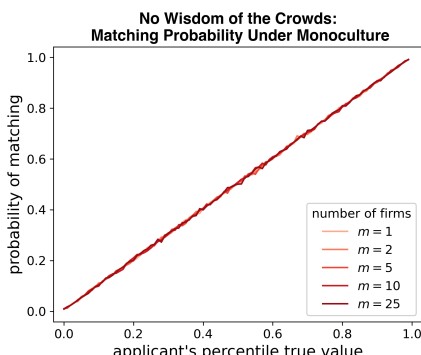

**Figure 2:** As the number of firms $m$ increases, $\Pr[\mu_{\mathrm{poly}}(v) \in \mathcal{F}]$ approaches a step function, meaning that only the highest-value applicants match. Meanwhile, $\Pr[\mu_{\mathrm{mono}}(v) \in \mathcal{F}]$ remains the same regardless of $m$; in particular, high-value applicants have positive probability of not matching. There are 1000 total applicants and firms have total capacity 500. Applicants have values drawn uniformly from $[0, 1]$ and uniformly-random preferences. Noise is drawn uniformly from $[-0.5, 0.5]$. Plot displays average over 1000 simulations.

Despite individual firms making noisy decisions, under polyculture, they collectively admit only the highest-value applicants as $m$ grows large (as illustrated in Figure 2). This gives a "wisdom of the crowds"-like result. Notably, firms do not directly share information with one another; rather, the market at large (specifically, the structure of stable matchings) facilitates the pooling of information.[5]

**Intuition for Theorem 1**  At a high level, the maximum estimated value—which is what determines if an applicant is matched—can effectively "distinguish" between applicants of higher and lower values. Consider applicants Alice and Bob. Alice's value $v_A$ is higher than Bob's value $v_B$. We show that if $\mathcal{D}$ is maximum-concentrating, the probability that Bob is matched but Alice is unmatched vanishes as the number of firms $m$ grows large. If Bob is matched but Alice is not, then Bob's maximum estimated value must be higher than Alice's. The probability this occurs is given by $\Pr[v_A + X_A^{(m)} < v_B + X_B^{(m)}]$, where $X_A^{(m)}$ and $X_B^{(m)}$ are independent variables distributed according to $X^{(m)}$. Since $\mathcal{D}$ is maximum concentrating, the probability that two independent draws from $X^{(m)}$ differ by more than the constant $v_A - v_B$ vanishes as $m$ grows large. Thus, Alice's maximum estimated value will almost always be higher than Bob's.

## 4.2  Applicant welfare

We now consider matching from the applicant's perspective. Our main result (Theorem 2) shows a few things. All applicants are more likely to be matched to their top choice firm under monoculture than under polyculture (part (i))—moreover, under monoculture, all applicants who are matched to *some* firm are matched to their *top* firm (part (ii)). This implies that total applicant welfare—as measured by the average rank of where applicants are matched—is optimal under monoculture (and better than under polyculture). Meanwhile, in part (iii), we show that there exists a set of applicants of positive measure that is simultaneously more likely to match to their top choice under monoculture while less likely to match overall, which implies that the applicant's variance is higher under monoculture. For these applicants, neither monoculture or polyculture stochastically dominates the other.

Define $\mathrm{Rank}_{\theta(v)}(f) := |\{f' : f' \succeq^{\theta(v)} f\}|$ to be the rank of an applicant's match, with $\mathrm{Rank}_{\theta(v)}(\emptyset) = 0$. $\mathrm{Rank}_v(\mu(v))$ is a random variable giving the rank of the firm an applicant of value $v$ is matched to under $\mu$. Let $(X_-, X_+)$ be the support of noise distribution $\mathcal{D}$.

**Theorem 2** (Likelihood of Matching to Top Choice or At All). *The following hold:*

*(i) For all $v$, the probability an applicant matches to their top choice is at least as high under monoculture as under polyculture: i.e., $\Pr[\mathrm{Rank}_v(\mu_{\mathrm{mono}}(v)) = 1] \geq \Pr[\mathrm{Rank}_v(\mu_{\mathrm{poly}}(v)) = 1]$. The inequality is strict for all $v \in (P_{\mathrm{mono}} - X_+, P_{\mathrm{mono}} - X_-)$, a set of positive $\eta$-measure.*

---

[5]This holds even despite an apparent "winner's curse" [12, 42, 15]: firms who extend an offer to an applicant tend to have an upward biased estimate of the applicant's value.

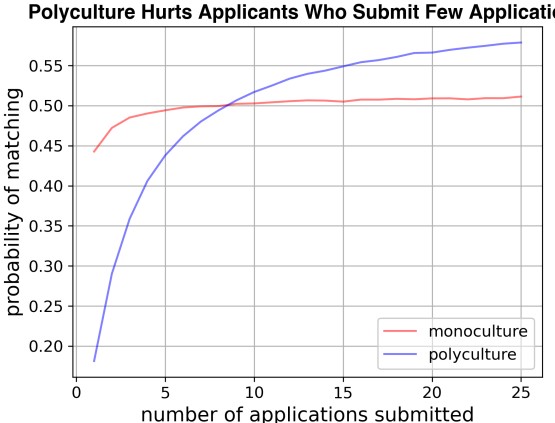

**Figure 3:** We plot the probability of matching conditional on $k$, the number of applications submitted. There are 1000 applicants and 25 firms have total capacity 500. Applicants have values drawn uniformly from $[0, 1]$ and uniformly-random preferences. Noise is drawn uniformly from $[-0.5, 0.5]$. Each applicant can apply to $k$ colleges, drawn uniformly from $\{1, 2, \cdots, 25\}$. Probabilities are averages over 10,000 simulations.

*(ii) For all $v$, if $\mu_{\mathrm{mono}}(v) \in \mathcal{F}$, then $\mathrm{Rank}_v(\mu_{\mathrm{mono}}(v)) = 1$, meaning that all applicants who match under monoculture are matched to their top choice, and that the matching attains optimal applicant welfare.*

*(iii) When $\mathcal{D}$ is maximum concentrating, for all $v \in (v_S, P_{\mathrm{mono}} - X_-)$, which is a set of positive $\eta$-measure, and for all $m$ sufficiently large, an applicant with value $v$ is less likely to match under monoculture than under polyculture: i.e., $\mathrm{Pr}[\mu_{\mathrm{mono}}(v) \in \mathcal{F}] < \mathrm{Pr}[\mu_{\mathrm{poly}}(v) \in \mathcal{F}]$.*

The high-level reasoning behind these results is fairly straightforward. First, in part (i), the probability an applicant is matched to their top choice is exactly the probability that their estimated value at that firm exceeds the shared cutoff: $P_{\mathrm{mono}}$ under monoculture and $P_{\mathrm{poly}}$ under polyculture. We know that $P_{\mathrm{mono}} < P_{\mathrm{poly}}$ (from Corollary 4), so this probability is at least as large under monoculture in comparison to under polyculture (and in many cases, strictly larger). To show part (ii), observe that if an applicant's common estimated value under monoculture exceeds the shared cutoff $P_{\mathrm{mono}}$, that applicant can afford any firm; thus, they are matched to their top choice. Finally, part (iii) is a consequence of Theorem 1, where we showed that for applicants with value $v$ above a certain threshold $v_S$, their likelihood of being matched approaches 1 as $m \to \infty$, while their likelihood of being matched remains constant (and less than 1) under monoculture when $v < P_{\mathrm{mono}} - X_-$.

### 4.3 An Extension: Differential Application Access.

We extend our primary setup to include *differential application access*—when different applicants submit a differing number of total applications. In practice, the number of applications submitted in college admissions varies significantly, and can be correlated with socioeconomic status: for example, in the 2021-2022 cycle, applicants who submitted 15 or more applications on the Common App were 2.5x as likely to have attended a private high school [34]. Our main result here (Theorem 3) shows that applicants who submit more applications do not benefit under monoculture, but do benefit under polyculture. Intuitively, this is because an applicant's estimated value is the same across firms under monoculture, so submitting applications does not increase the likelihood that an applicant will have a higher estimated value. Meanwhile, under polyculture, submitting more applications increases the likelihood that an applicant will receive a higher estimated value, since estimated values are drawn independently. This is shown in Figure 3. The full extended model and results are given in Appendix A.

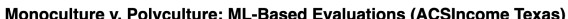

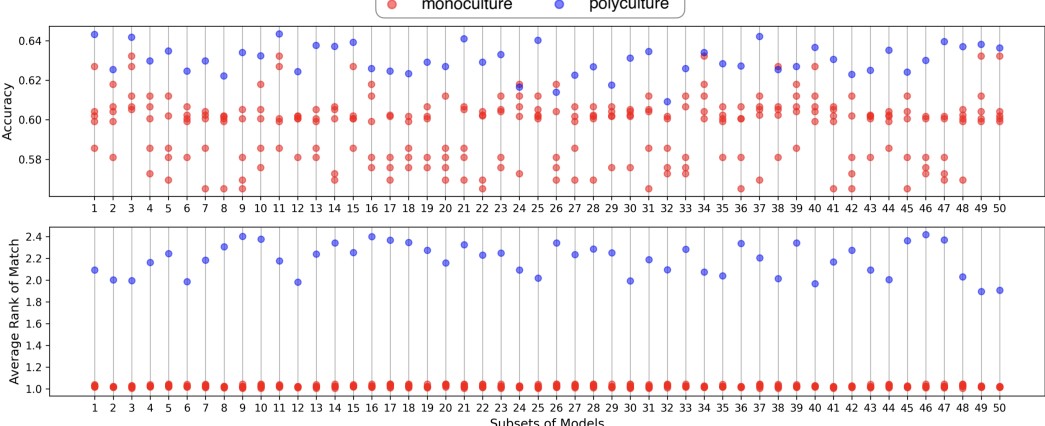

**Figure 4:** Each column corresponds to a random subset of 5 ML models. Red dots correspond to markets where all firms use the same (1 of the 5) ML models to rank applicants (monoculture). Blue dots correspond to markets where all firms use a different ML model to rank applicants (polyculture). The top figure illustrates the accuracy of each market (the percentage of matched applicants with a positive label); here, polyculture outperforms monoculture, as predicted by Theorem 1. The bottom figure illustrates the average rank choice of applicant matches; here, monoculture outperforms polyculture, as predicted by Theorem 2. See Section 5 for details.

## 5 Computational Experiments

We provide an overview of our computational experiments, with full details in Appendices C and D. Code to reproduce our experiments (and all other figures) is available at a publicly-available repository.[6] The code can also be adapted to test further variations in market settings.

**ML Simulations: ACSIncome** We test our theoretical predictions in a setting where preference approximation is done via ML models. Applicants have 0-1 true values, and firms rank applicants based on a prediction outputted by an ML model (following [11]). We consider models using different sets of features, and compare markets where firms use different models to evaluate applicants (polyculture) and markets where firms use the same model (monoculture). We use the ACSIncome dataset—specifically, individuals from Texas in 2018 [19]. (We also perform experiments using data from California; see Appendix C.) The ACSIncome dataset has 10 features that are used to predict a binary 0-1 outcome.[7] For each of the $\binom{10}{2} = 45$ pairs of features, we train a logistic regression model using a 90 percent train split. We then evaluate markets where each applicant corresponds to a point in the test split. This yields markets with 13,593 total applicants, of which 5,055 have value 1 and 8,538 have value 0. We assume that there are 5 firms, each with capacity 1000, and that applicants have uniformly random preferences over these firms. Each of the 45 models yields a monoculture market in which all 5 firms use that same model to rank applicants. We let the *accuracy* of a single model be the average value of matched applicants in the corresponding monoculture market (i.e., the percentage of matched applicants with value 1). We then focus our analysis on the 20 highest-accuracy models (roughly speaking, the models that firms may consider using).

From these 20 models, we randomly select 50 subsets of 5 models that differ by at most 0.05 in accuracy, producing subsets of "candidate models." (If the range of accuracies is too large, then this means that one model may be clearly superior than the others.) For each subset, we consider the 5 corresponding monoculture markets (where each firm uses the same model to evaluate applicants), and 1 corresponding polyculture market (where each firm uses a different model). For each market, we plot (1) the average value of matched applicants (i.e., accuracy), and (2) the average firm rank among matched applicants (testing Theorems 1 and 2 respectively). These plots are given in Figure 4, where each column corresponds to one subset of 5 models. As Theorems 1 and 2 predict, polyculture outperform monoculture in terms of accuracy, but monoculture outperforms polyculture in terms of

---

[6] https://github.com/kennylpeng/monoculture

[7] In particular, the task is to predict whether or not a person's income is above a threshold.

rank of applicant matches. Indeed, for 47 out of 50 subsets of models, the polyculture market is more accurate than *all* monoculture markets (this effect is present but less strong for California[8]); for 50 out of 50 subsets of models, the monoculture market produces better average applicant outcomes (indeed, matched applicants almost always match to their top choice). We also test theoretical predictions of Theorem 3 under this setup, detailed in Appendix C. Further experiments in which we varied applicant preferences and firm capacities yielded similar qualitative results.

**Correlated Applicant Preferences**   Our theoretical results are in a setting where applicants have uniformly random preferences over firms—i.e., no firm is on average more desirable than any other. While this assumption is common in the matching literature (e.g., much of the work on single versus multiple tie-breaking) for theoretical tractability, one concern is that our insights do not extend beyond this assumption. Here, we test the predictions of our main results in broader settings.

We consider computational experiments in which applicants have correlated preferences over firms. To introduce correlation, we follow Ashlagi et al. [8] in using a random utility model adopted from Hitsch et al. [26], where each applicant additional has a "location" parameter drawn uniform at random, and each firm has a "vertical quality" and a "location" parameter. Setup details and experimental results are given in Appendix D. Simulations show that the results extend to the full range of component weights considered. In particular, Figures 7 to 10 demonstrate that the (directional) results of Theorems 1, 2 and 3 are robust to correlation in applicant preferences. Experiments in which we further varied firm capacities revealed similar qualitative results.

## 6   Conclusion

In this paper, we studied the effects of algorithmic monoculture by employing a matching markets model. This allowed us to capture a key feature of domains that may be affected by monoculture: competition between and the limited capacity of decision-makers (firms, colleges). By incorporating these features, and by considering large markets with many participants, we strengthened, challenged, and expanded upon the existing understanding of monoculture. We presented three main theoretical findings, which we verified in computational experiments. All else equal, monoculture (1) selects less-preferred applicants, (2) yields higher overall applicant welfare, though effects for individual applicants depend on their value to decision-makers and risk tolerance, and (3) is more robust to differences in the number of applications submitted by different applicants.

**Limitations**   The theoretical results we presented are derived in a setting with strong assumptions. The first assumption is of homogeneous firms—that students have uniformly random preferences over firms and that firms have equal capacities. The intuition behind the theoretical results suggest robustness to this assumption—a claim supported by computational experiments, which consider varying structures of correlation in student preferences (as well as estimated values derived from ML models rather than i.i.d. noise). A second assumption is of maximum-concentrating noise. In comparison to the homogeneous firms assumption, it should be emphasized that Theorem 1 does not necessarily generalize beyond this assumption, as suggested by some of our empirical results (see Appendix C for ML experiments for the California subset of the data). In fact, our follow-up work demonstrates that long-tailed noise results in the *opposite* finding in Theorem 1 [40]. More broadly, the model we present does not consider other relevant aspects of matching markets—such as biases in evaluations, heterogeneity across different groups of applicants, and search frictions. Studying monoculture in these broader settings is an interesting question for future work.

**Broader Impacts**   Our work analyzes the emerging concern of algorithmic monoculture, in which many decision-makers rely on the same algorithm to evaluate individuals. Our work aims to build a stronger understanding of the potential risks of algorithmic monoculture, with the goal of creating a positive societal impact by informing potential regulatory decisions. Our work is not comprehensive; in particular, our analysis focuses on quantitative measures of welfare, and does not consider rights- or claims-based approaches (see [32]) that may be especially relevant in high-stakes settings.

---

[8]In California, for 19 out of 50 subsets of models, the polyculture market is more accurate than *all* monoculture markets and for 44 out of 50 subsets of models, the polyculture market outperforms the *average* of the 5 monoculture markets. This suggests that a "weak" version of Theorem 1 holds broadly. Importantly, this may due to a lack of strict adherence to the assumption that noise is max-concentrating, which Theorem 1 requires.

## Acknowledgments

We especially thank S. Nageeb Ali, Nick Arnosti, Rishi Bommasani, Kathleen Creel, Sophie Greenwood, Shomik Jain, Jon Kleinberg, Rajiv Movva, Manish Raghavan, Ran Shorrer, and Ashia Wilson for helpful feedback and discussion. NG also thanks Itai Ashlagi for explaining school matching tiebreaking to him many years ago. We also thank participants of the Digital Life Initiative seminar, the Marketplace Innovation Workshop, and the ESIF Economics and AI+ML Meeting for thoughtful questions and feedback. KP is supported by the Digital Life Initiative fellowship. NG is supported by NSF CAREER IIS-2339427, and Cornell Tech Urban Tech Hub, Meta, and Amazon research awards.

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

# A  An Extension: Differential Application Access

We now consider when different applicants can apply to different numbers of firms (analogously, colleges)—which we refer to as *differential application access*. In particular, we assume some discrete probability distribution $\kappa$ over $\{1, 2, \cdots, m\}$ that determines the number of firms an applicant can apply to. In particular, an applicant with value $v$ can apply to $k(v)$ firms, where $k(v)$ is drawn independently from $\kappa$. All other aspects of our model remain the same.

Under both monoculture and polyculture there is a natural Nash equilibrium in which applicants all apply to their top $k$ firms. In this equilibrium, we find that applicants who can apply to more firms benefit under polyculture, producing inequitable outcomes. This is not true under monoculture. Consequently, while firm welfare is not harmed when applicants can apply to a different number of firms under monoculture, it is harmed under polyculture. Together, these results suggest that monoculture is more robust than polyculture with respect to differential application access.

## A.1  Model setup

We begin by instantiating differential application access within our model. An applicant $v$ who can apply to $k$ firms must choose an application strategy (i.e., which $k$ firms to apply to). A strategy profile is given by a vector of maps $S = (S_1, S_2, \cdots, S_m)$, where $S_k : \mathbb{R} \times \mathcal{R} \to \{S \in 2^{\mathcal{F}} : |S| = k\}$. Therefore, an applicant who who has value $v$ and a preference ordering $\succ$ and who can apply to $k$ firms applies to the firms in $S_k(v, \succ)$. Therefore, a strategy profile uniquely determines where an applicant with a given value, preference list, and level of application access applies.

We will focus on the strategy profile $S$ such that $S_k(v, \succ)$ gives the $k$ most preferred firms according to $\succ$. Therefore, strategies do not depend at all on $v$. We will later show that this intuitive set of strategies forms a Nash equilibrium. Fixing this strategy profile $S$, we can now define $\theta(v)$ for monoculture and polyculture in the presence of differential application access.

**Monoculture with differential application access.**   Define

$$\theta_{\mathrm{mono},\kappa}(v) = (\succ, e) \tag{10}$$

where

$$e_c = \begin{cases} v + X & f \in S_{k(v)}(v, \succ) \\ -\infty & f \notin S_{k(v)}(v, \succ) \end{cases}, \tag{11}$$

for $\succ$ drawn uniformly at random from $\mathcal{R}$, $k(v)$ drawn from $\kappa$, and $X$ drawn from $\mathcal{D}$. As before, an applicant with value $v$ is given a random preference ordering over firms. However, now the applicant's estimated values across firms depends also on a random draw from $\kappa$, which determines how many firms they can apply to. Recall that $e$ gives the vector of estimated values of an applicant across firms. Therefore, under monoculture, this estimated value is shared across the firms the applicant applies to (which are the applicant's top-$k(v)$ most preferred firms), but is $-\infty$ at other firms (meaning that the applicant will never be matched to firms they do not apply to).

**Polyculture with differential application access.**   Define

$$\theta_{\mathrm{poly},\kappa}(v) = (\succ, e) \tag{12}$$

where

$$e_f = \begin{cases} v + X_f & f \in S_{k(v)}(v, \succ) \\ -\infty & f \notin S_{k(v)}(v, \succ) \end{cases}, \tag{13}$$

for $\succ$ drawn uniformly at random from $\mathcal{R}$, $k(v)$ drawn from $\kappa$, and $X_1, \cdots, X_m \overset{\text{iid}}{\sim} \mathcal{D}$. Here, an applicant gets an independently drawn estimated value at each firm they apply to.

We now give the Equal Cutoffs Lemma in this setting, which again, is a consequence of symmetry in applicants' preferences over firms.

**Lemma 5** (Equal Cutoffs Lemma for Differential Application Access). *If $\theta \in \{\theta_{\mathrm{mono},\kappa}, \theta_{\mathrm{poly},\kappa}\}$, then there is a unique vector of market-clearing cutoffs $P$, and $P_1 = P_2 = \cdots = P_m$.*

We denote these unique stable matchings by $\mu_{\mathrm{mono},\kappa}$ and $\mu_{\mathrm{poly},\kappa}$, with corresponding *shared cutoffs* $P_{\mathrm{mono},\kappa}$ and $P_{\mathrm{poly},\kappa}$. Recall that under our abuse of notation, $\mu_{\mathrm{mono},\kappa}(v) := \mu_{\mathrm{mono},\kappa}(\theta_{\mathrm{mono},\kappa}(v))$ and $\mu_{\mathrm{poly},\kappa}(v) := \mu_{\mathrm{poly},\kappa}(\theta_{\mathrm{poly},\kappa}(v))$ are the random variables representing where $v$ is matched under monoculture and polyculture.

The strategy profile we consider here forms an ex-ante Nash equilibrium, meaning that no applicants—prior to knowing their estimated values at each firm—would apply to a set of firms different from their top $k$.

**Proposition 6** (Nash Equilibrium). *For any $\kappa$, and under both monoculture and polyculture, no applicant benefits ex-ante by deviating from $S$.*

This is an immediate consequence of Lemma 5. Since cutoffs at all firms are the same, an applicant is strictly harmed by applying to a set of firms other than their top $k$, since doing so does not increase their chances of being matched to any given firm (or set of firms).

### A.2 Analysis

We now show that while monoculture is robust to differential application access, polyculture is not. Our main result in this section is that under monoculture, applicants who apply to more firms do not gain an advantage, while under polyculture, applicants who apply to more firms are more likely to be matched. Again recall that $(X_-, X_+)$ is the interval on which $\mathcal{D}$ is supported.

**Theorem 3** (Differential Application Access in Monoculture and Polyculture). *The following hold:*

(i) *In monoculture, applicants who apply to more firms do not gain an advantage: For all $v$,*

$$\Pr[\mu_{\mathrm{mono},\kappa}(v) \in \mathcal{F} \mid k(v) = k] = \Pr[\mu_{\mathrm{mono}}(v) \in \mathcal{F}] \qquad (14)$$

*is constant in $k$.*

(ii) *In polyculture, applicants who apply to more firms gain an advantage: For all $v$,*

$$\Pr[\mu_{\mathrm{poly},\kappa}(v) \in \mathcal{F} \mid k(v) = k] \qquad (15)$$

*is increasing in $k$, and strictly increasing in $k$ for $v \in (P_{\mathrm{poly},\kappa} - X_+, P_{\mathrm{poly},\kappa} - X_-)$.*

This result is illustrated in Figure 3, which shows that while the probability of matching does not depend significantly on $k$ under monoculture, it is increasing in $k$ under polyculture. A consequence of the result is that students with higher values can go unmatched while students with lower values but who submit more applications may be matched. From the firm (or college) perspective, this means that the ability to distinguish higher- and lower-value applicants is diminished. We demonstrate this in our computational experiments (see, e.g., Figure 10). This further suggests that the strong result of Theorem 1 is not robust to differential application access, and as a consequence, firms and colleges under polyculture are incentivized to "level the playing field" to ensure that applicants each submit a similar number of total applications.

Notice that, as in Theorem 1, the results here hold independently of the relationship between the noise distributions in monoculture and polyculture.

## B  Extended Related Work

Here, we further detail our relationship to the algorithmic monoculture and matching markets literature. We conclude with a discussion of how our work bridges these two lines of work, and reconciles some competing intuitions.

**Algorithmic monoculture.**  Closest to our work is that of Kleinberg and Raghavan [35], who introduce a model of algorithmic monoculture in which there are two decision-makers who each hire one applicant. As in our model, decision-makers have shared true preferences over a set of applicants, instantiated as a ranked list. However, each decision-maker only has access to a noisy version of this ranked list: they each can either use an idiosyncratic noisy list or a common (but more accurate) noisy list. One decision-maker then selects the top applicant on their list. The second decision-maker chooses the top applicant on their list that has not yet been chosen. Welfare is measured by the

rank of the chosen applicants according to the true preference list. Kleinberg and Raghavan [35] show that the decision-makers can be better off if they both use their idiosyncratic (more noisy) lists in comparison to using the shared list. In our work, we show that this effect becomes more pronounced when considering a large market with many decision-makers and applicants. By building on the model of Azevedo and Leshno [10], we overcome issues of tractability, while also gaining a cleaner interpretation for why the effect found in Kleinberg and Raghavan [35] holds, in terms of distributional order statistics and their effect on admission thresholds for each decision-maker. Moreover, we are able to model two-sided effects, where applicants also have preferences over decision-makers. Absent in our work is the game-theoretic analysis of Kleinberg and Raghavan [35], who show that there are cases where both decision-makers use the shared list in the Nash equilibrium, even when both would be better off using idiosyncratic lists.

While Kleinberg and Raghavan [35] focus on the welfare of decision-makers, Bommasani et al. [11] consider the effect of algorithmic monoculture on applicants. In their setup, applicants are subject to binary decisions from several decision-makers. They define the rate of systemic failure to be the probability that a randomly drawn applicant receives a negative decision from all decision-makers. In computational experiments, they provide evidence that when decision-makers train machine learning models using the same data, the rate of systemic failure tends to increase. Toups et al. [43] document homogeneous outcomes in several commercial APIs, which they suggest could be a consequence of algorithmic monoculture arising from shared data or model usage. These empirical studies do not consider market effects such as fixed capacity constraints of decision-makers; in the setup we consider, since decision-makers have fixed capacity, the number of applicants who experience systemic failure remains constant (assuming that applicants too only "accept" one offer).

Indeed, decision-maker capacity constraints play an important part in our results. Jagadeesan et al. [30] consider a different structure, in which several "model-providers" compete over users. In their setting, a user (noisily) prefers model-providers that produce a more accurate prediction for the user. A key difference between this model and ours is that the analog to the decision-maker (the model provider) does not have limited capacity; rather, a single model-provider can serve an unlimited number of users. Moreover, the model provider does not seek to evaluate users, but rather to make predictions in their service. Jagadeesan et al. [30] shows that as the ability of model-providers to produce more accurate predictions increases overall, the social welfare of users can decrease as a consequence of growing homogeneity in predictions. (Users benefit from diversity since it means that there is likely at least one model provider that is accurate for them.) We also note that in Jagadeesan et al. [30]'s model, monoculture arises endogenously, while monoculture is specified exogenously in our model.

An emerging line of work also considers normative aspects of monoculture. Creel and Hellman [16] argue that *arbitrariness* in algorithmic decision-making is not generally of moral concern—however, they argue, arbitrary decision-making *at scale* is of concern, since it can result in *systemic exclusion*: "Exclusion from a broad swath of opportunity in an important sector of life is likely to be morally problematic." Our work challenges the implicit assumption that the widespread adoption of a common algorithm necessarily increases the number of individuals excluded from all opportunities. When the number of "opportunities" (positions at a firm, seats at a college) is constant, the number of individuals that receive these opportunities (and who are denied from all opportunities) should remain constant. Of course, not all domains of algorithmic decision-making share this feature. In the domains that do, our work suggests systemic exclusion is not clearly the primary concern, but rather *who* faces systemic exclusion, as well as the other measures of welfare (e.g., worker power). Jain et al. [31] argue that algorithmic decision-making should be viewed through the lens of *opportunity pluralism* [22]. A key implication of their argument is that the use of algorithmic decision-making by one or multiple decision-makers should be analyzed with respect to its impact on how opportunities are restricted at large. (One possible intervention is to randomize outcomes, as argued for by Jain et al. [32].) Our work contributes to this type of analysis, helping develop a more precise understanding of the effects of algorithmic decision-making at a system level.

**Matching markets.**  The model we introduce here builds on the broader analysis of matching markets. In particular, a substantial literature analyzes the outcomes of random matching markets. Theoretical results have established the expected number of stable matchings (i.e., the size of the core) as well as the distribution of outcomes for each participant (e.g., Pittel [41], Immorlica and Mahdian [27], Ashlagi et al. [8]).

We depart from this literature by modeling the process of *preference approximation*. Whereas existing analysis focuses on a single set of stated preferences, we assume that participants (specifically, decision-makers) can only obtain or communicate an *approximation* of their own preferences. As a result, stable matchings are computed with respect to these approximate preferences. We then analyze the matching outcome with respect to the *true* preferences of participants. Our model can thus be viewed as having two components:

- First, a *preference approximation* model, in which each participant acquires approximate knowledge of their true preferences over the other side.[9]

- Second, a *stable matching* model, which takes as input the approximate preferences of participants, and outputs a stable matching with respect to these preferences.

For the stable matching model, we adopt a model with a continuum of students [3, 10]. This allows for an especially tractable analysis. A limitation of this approach is that we do not capture the effects of stochasticity in small markets (particularly, where the capacity of decision-makers is small). In our framework, it is possible to substitute a stable matching model that is more amenable to such analysis, such as the standard finite model or that of Arnosti [6].

An emerging line of work similarly considers matching markets (both centralized and decentralized) in which participants are unsure of their true preferences or broader information about the market. Existing work primarily focuses on settings where participants gain information sequentially (e.g., Liu et al. [36, 37], Dai and Jordan [17], Ionescu et al. [29], Jeloudar et al. [33], Immorlica et al. [28]). Our work focuses instead on a one-shot setting, where we analyze the outcomes of matchings that arise given the current information of each participant.

Closest to our work is a model introduced by Castera et al. [13] to analyze statistical discrimination[10] in stable matching. In their model, the values of applicants from two groups are estimated by two decision-makers. Like us, they leverage a continuum model of applicants. Their findings focus on aggregate measures of applicant-side welfare across groups. They show, for example, that the group of applicants evaluated with less correlation is matched with higher probability. More directly related to our work, they show that decreased correlation (even in one group), reduces the number of applicants who receive their top choice; this parallels our finding in Theorem 2(ii), which implies that fewer applicants are matched to their top choice under polyculture (i.e., when there is total correlation). This finding reflects a general result developed in the school choice literature: that increasing correlation in how decision-makers evaluate applicants also increases the total number of applicants matched to their top choice. We discuss this line of literature next.

**Tie-breaking rules in school choice.** Our work has close connections to the matching markets literature studying tie-breaking rules in the context of school choice, where lottery numbers are often used to determine the school preferences inputted into stable matching mechanisms [1]. A significant line of empirical and theoretical work compares the use of *single tie-breaking* (STB) where each student is assigned a single lottery number across all schools, and *multiple tie-breaking* (MTB) where each student is assigned a different random lottery number for each school.

A main finding across this literature is that STB results in more students being assigned to their top choice school (e.g., Abdulkadiroğlu et al. [2], De Haan et al. [18], Ashlagi et al. [9], Arnosti [7], Allman et al. [5]). This finding is echoed in our Theorem 2. Indeed, one may directly compare monoculture to STB, since each applicant in our model has a single estimated value used by all decision-makers. From the perspective of population-level statistics of overall student welfare, monoculture and STB are both equivalent to a model in which all firms (schools) have the same preferences over applicants. At a high level, both sets of results rely on a similar intuition: in settings

---

[9]Note that in our model, only the preferences of decision-makers are "approximated," while applicants are assumed to know their own true preferences. This preference approximation further relates to recent literature in modeling the role of standardized testing in admissions, where tests (among other features) help schools estimate their value for each applicant Emelianov et al. [20], Garg et al. [23], Liu and Garg [38], Niu et al. [39]. However, studying more than two schools in this framework has proven theoretically challenging. Castera et al. [13] analyzes a related model to understand the effect of statistical discrimination in matching; they too focus on a two school setting. A broader insight of our analysis is that the limiting regime in which the number of schools grows large can be more tractable than the strictly finite setting.

[10]See Fang and Moro [21] for a survey of the statistical discrimination literature.

with two-sided preferences, applicants as a whole benefit if *their* preferences most influence matches; this happens when schools share preferences (as induced by either monoculture or STB, but not by polyculture or MTB).

Our setup departs from the tie-breaking literature by having each applicant's estimate depend on the applicant's *true value*. Thus, the "noise" in their estimate (the equivalent of a tie-breaking score) is not just differentiating between equivalent applicants; it can move a lower-value applicant over a higher-value one. As a result, our analysis further focuses on comparing monoculture and polyculture from the perspective of an *individual* applicant with a particular value. This also allows us to analyze welfare from the perspective of the firm (who prefer to be matched with applicants of higher true value). Polyculture and MTB also share differences even at the population level; in polyculture, there remains some correlation between "lottery numbers" across schools (firms) depending on the applicant's true value.

In addition to showing that STB results in more students being matched to their top choice,[11] Arnosti [7] shows that students who list many schools are more likely to be matched under MTB, while students who list few schools are more likely to be matched under STB. This accords with our finding in Theorem 3 that applicants who submit more applications gain an advantage under polyculture but not under monoculture. We show that this further implies that (1) students with lower true values but who submit more applications can be matched over students with higher true values but who submit fewer applications, and that (2) differences in the number of applications submitted by students reduces firm welfare under polyculture, but not under monoculture.

We anticipate that the techniques developed in the study of tie-breaking rules may be useful for analyzing outcomes in algorithmic monoculture, especially on the applicant side. The introduction of heterogeneity on the applicant side adds a dimension of analysis beyond what is considered by the existing tie-breaking literature, and which is ripe for further inquiry.

**Bridging the literatures.** Our work bridges two lines of work: algorithmic monoculture and matching markets. We briefly elaborate on the nature of this connection. The initial inquiry into algorithmic monoculture has been limited by a lack of market-level analysis. The choice of a firm to adopt a certain hiring algorithm not only affects outcomes at that firm, but also other firms. The nature of this effect is further dictated by the preferences of applicants. We have shown how machinery in the matching markets literature is well-equipped to handle these effects. Adopting this machinery enhances our understanding of algorithmic monoculture. While the algorithmic monoculture literature has largely pointed to negative effects of monoculture, both on the firm and applicant side, we showed that monoculture can have positive welfare effects for applicants—a competing intuition that can be found in the matching markets literature on tie-breaking, which has found that correlation in school-side preferences can benefit students in a school choice setting. In this way, we anticipate that the matching markets literature can provide further insight into the emerging study of algorithmic monoculture.

At the same time, our work expands the scope of questions that can be analyzed using a matching markets framework. While existing work in matching markets largely takes preferences at face value, we further model a stage of *preference approximation*, in which participants form noisy approximations of their true preferences. In this work, we focused on correlation in this approximation process, but the question can be posed more generally. Our results suggest that the structure of the approximation process can have intriguing consequences on the quality of the resulting matching.

## C   Computational Experiments: ML-Based Evaluations

**Details of Data and Experimental Setup**   We consider the ACSIncome dataset [19], which consists of predicting whether an individual makes more than 50,000 dollars based on 10 features, including demographic features. Importantly, this is not a legitimate setup for making any assessment of an individual's employability, but rather intended as a toy task in which to demonstrate how our theoretical setup can transfer to a ML-based setup. The data is available via the Python package `folktables`.[12] The package is released under an MIT license. The data may be restricted to

---

[11]In fact, Arnosti [7] shows more precise results, characterizing the probability that a student is matched to one of their top $k$ choices.

[12]https://github.com/socialfoundations/folktables

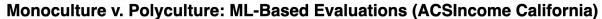

**Monoculture v. Polyculture: ML-Based Evaluations (ACSIncome California)**

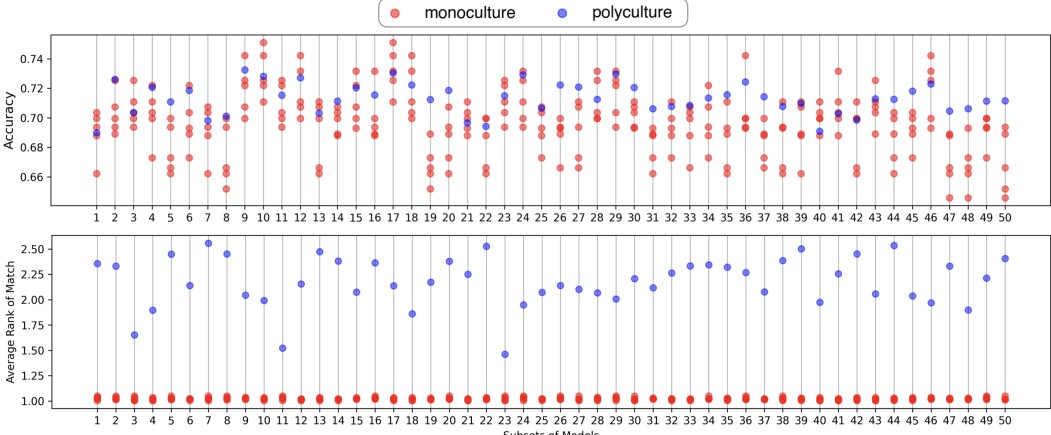

**Figure 5:** Each column corresponds to a random subset of 5 ML models. Red dots correspond to markets where all firms use the same (1 of the 5) ML models to rank applicants (monoculture). Blue dots correspond to markets where all firms use a different ML model to rank applicants (polyculture). The top figure illustrates the accuracy of each market (the percentage of matched applicants with a positive label); here, polyculture outperforms monoculture, as predicted by Theorem 1. The bottom figure illustrates the average rank choice of applicant matches; here, monoculture outperforms polyculture, as predicted by Theorem 2. See Appendix C for details.

particular states and years. We focus on two subsets of the data: Texas in 2018 and California in 2018, two subsets which are analyzed by [19] (among many other subsets).

The models we train use the default `sklearn Logistic Regression` hyperparameters. Additional optimization was not necessary for our purposes, since it was not important to improve upon the accuracy of individual models, but rather to understand matching markets in which ML models are used to form preferences. All experiments can be completed on a typical laptop in at most a few hours.

**Results for California**  In addition to the Texas 2018 data, we perform the same experiments on the California 2018 data. Again, for each of the $\binom{10}{2} = 45$ pairs of features, we train a logistic regression model using a 90 percent train split. We then evaluate markets where each applicant corresponds to a point in the test split. This yields markets with 19,567 total applicants, of which 7,933 have value 1 and 11,634 have value 0. We assume that there are 5 firms, each with capacity 1000, and that applicants have uniformly random preferences over these firms. Each of the 45 models yields a monoculture market in which all 5 firms use that same model to rank applicants. We let the *accuracy* of a single model be the average value of matched applicants in the corresponding monoculture market. This gives the percentage of matched applicants with value 1 in a market where all firms used that model. We then focus our analysis on the 20 highest-accuracy models (roughly speaking, the models that firms may consider using).

From these 20 models, we randomly select 50 subsets of 5 models that differ by at most 0.05 in accuracy, producing subsets of "candidate models." (If the range of accuracies is too large, then this means that one model may be clearly superior than the others.) For each subset, we consider the 5 corresponding monoculture markets (where each firm uses the same model to evaluate applicants), and 1 corresponding polyculture market (where each firm uses a different model). For each market, we plot (1) the average value of matched applicants, and (2) the average firm rank among matched applicants (testing Theorems 1 and 2 respectively). These plots are given in Figure 4, where each column corresponds to one subset of 5 models. As Theorems 1 and 2 predict, polyculture outperform monoculture in terms of accuracy, but monoculture outperforms polyculture in terms of rank of applicant matches. Indeed, for 19 out of 50 subsets of models, the polyculture market is more accurate than *all* monoculture markets and for 44 out of 50 subsets of models, the polyculture market outperforms the *average* of the 5 monoculture markets. This suggests that a "weak" version of Theorem 1 holds broadly. Again, for 50 out of 50 subsets of models, the monoculture market

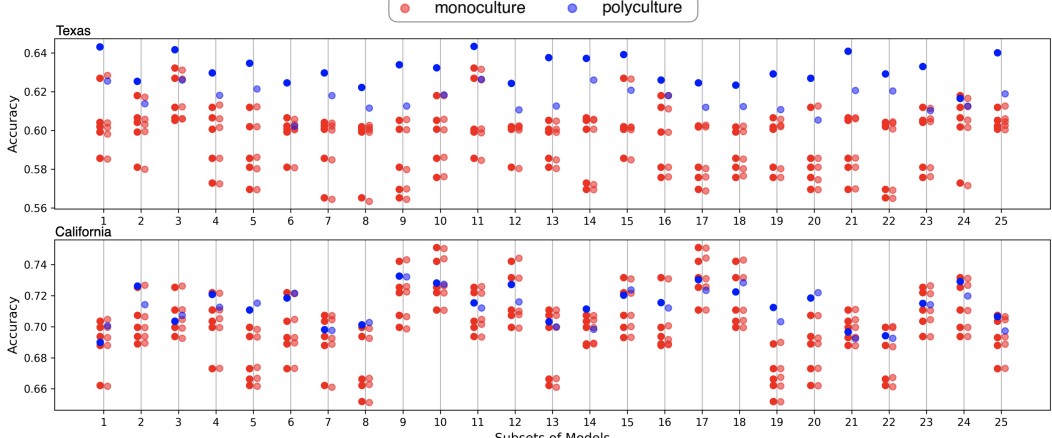

**Figure 6:** Each column corresponds to a random subset of 5 ML models. Red dots correspond to markets where all firms use the same (1 of the 5) ML models to rank applicants (monoculture). Blue dots correspond to markets where all firms use a different ML model to rank applicants (polyculture). Both figures illustrate the accuracy of each market (the percentage of matched applicants with a positive label). The top figure gives the Texas data, and the bottom figure gives the California data. Dots on the left side of each column correspond to no differential application access, while the dots on the right side of each column correspond to differential application access. One may observe that the blue dots change more from the left to the right side of each column, indicating that polyculture is less robust to differential application access.

produces better average applicant outcomes (indeed, matched applicants almost always match to their top choice).

**Testing Differential Application Access (Theorem 3)**    Using the same setup, we further test the prediction of Theorem 3, that monoculture is more robust to differential application access than polyculture. Here, we allow each applicant to apply to $k$ firms, where $k$ is drawn uniformly at random from $\{1, 2, 3, 4, 5\}$. Proposition 6 implies that (given the symmetric setup we consider here) the optimal strategy for an applicant who applies to $k$ firms is to apply to their $k$ most preferred firms. We then test the robustness of the accuracy of the markets considered above under this form of differential application access.

For the Texas data, the accuracy of polyculture markets has an average absolute change (between no differential application access and differential application access) of 1.54%, while the average absolute change for monoculture markets is 0.056%. For California, these two figures are 0.58% and 0.061% respectively. This is consistent with the prediction of Theorem 3 that monoculture is more robust under differential application access than polyculture.

## D   Computational Experiments: Correlated Preferences

In this section, we perform computational experiments to test the accuracy of predictions from our theoretical results in more general settings. We focus on our three broad theoretical findings: that all else equal, in comparison to polyculture, monoculture:

   (1)  selects less-preferred applicants (Theorem 1),

   (2)  matches more applicants to their top choice, though effects for individual applicants vary depending on their value (Theorem 2),

   (3)  is more robust to differential application access (Theorem 3).

All experiments below can be completed on a typical laptop in at most a few hours.

**Correlated applicant preferences.**    We proved our theoretical results in a setting where applicants have uniformly random preferences over firms. We consider computational experiments in which

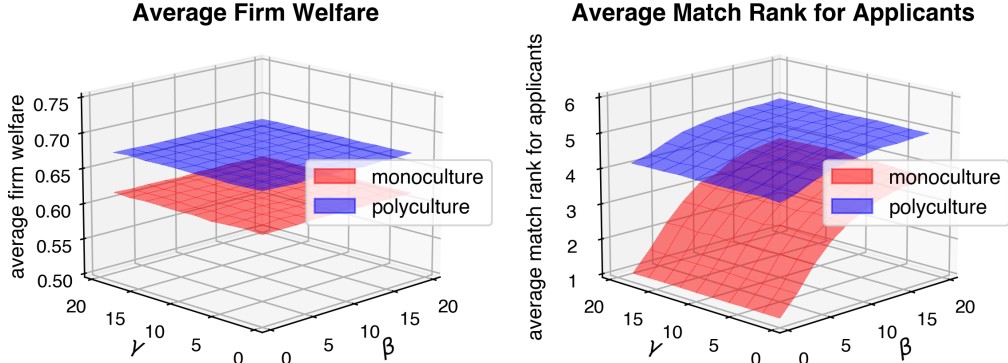

**Figure 7:** The average percentile value of matched applicants (left) and the average rank of firms matched to applicants (right), as a function of $\beta$ and $\gamma$, which control the level of global and local correlation between applicant preferences. (The case $\beta = \gamma = 0$ corresponds to our theoretical setup.) For all choices of $\beta$ and $\gamma$ we consider here, average firm welfare is higher under polyculture while average applicant welfare is higher under monoculture (note that lower corresponds to a better outcome in the right plot).

applicants can have correlated preferences over firms. To introduce this correlation, we follow [8] in using a random utility model adopted from [26].[13]

The model generates applicant preferences in the following way. Each applicant $i$ has a characteristic $x_i^D$ drawn independently from $U[0, 1]$. Each firm has two characteristics, $x_f^A$ and $x_f^D$, both drawn independently from $U[0, 1]$. Then the utility applicant $i$ for being matched to firm $f$ is

$$u_i(f) = \beta x_f^A - \gamma(x_i^D - x_f^D)^2 + \varepsilon_{if}, \tag{16}$$

where $\varepsilon_{if}$ is drawn independently from the standard logistic distribution. Here, $x_f^A$ is a vertical measure of firm quality, shared by all applicants, so $\beta$ controls the level of correlation between preferences. $x_i^D$ and $x_f^D$ are "locations" of the applicant and firm, and $\gamma$ controls the preference for an applicant to be "close" to the firm. $\varepsilon_{if}$ accounts for other idiosyncratic factors.

When $\beta = \gamma = 0$, we recover the uniformly random preferences used in our theoretical analysis. When $\beta$ grows large, preferences become fully correlated: applicants all have the same preferences. When $\gamma$ increases, there is more correlation in preferences between "nearby" applicants.

**Experiment details.** We now consider a market in which there are 1000 applicants and 10 firms each with capacity 50. (So half of applicants are matched.) We let applicant values be distributed uniformly from 0 to 1 (so $\eta$ is the uniform measure on $[0, 1]$), and take the noise distribution $\mathcal{D}$ to be the Gaussian distribution $\mathcal{N}(0, \frac{1}{2})$. We vary $\beta$ and $\gamma$ between 0 and 20. The plots we show are all in this setting, and provide averages over 100 random instantiations of the market.

### D.1 Firm welfare

We test our first prediction: that polyculture selects more-preferred applicants in comparison to monoculture. To do this, we compute the average percentile value of applicants matched to firms, so that a higher average percentile corresponds to higher firm welfare. As shown in Figure 7 (left), average firm welfare is higher under polyculture across all $\beta$ and $\gamma$ we consider. In fact, our experiments suggest that firm welfare in both monoculture and polyculture remains consistent regardless of the correlation structure we choose.

### D.2 Applicant welfare

We now test our second prediction: that monoculture yields higher total applicant welfare. Here, we measure total applicant welfare as the average rank of applicant matches—conditional on matching

---

[13]In [8], the model is used to generate preferences on both sides of the market; in our setting, we only use the model to generate applicant-side preferences, i.e., how applicants rank firms.

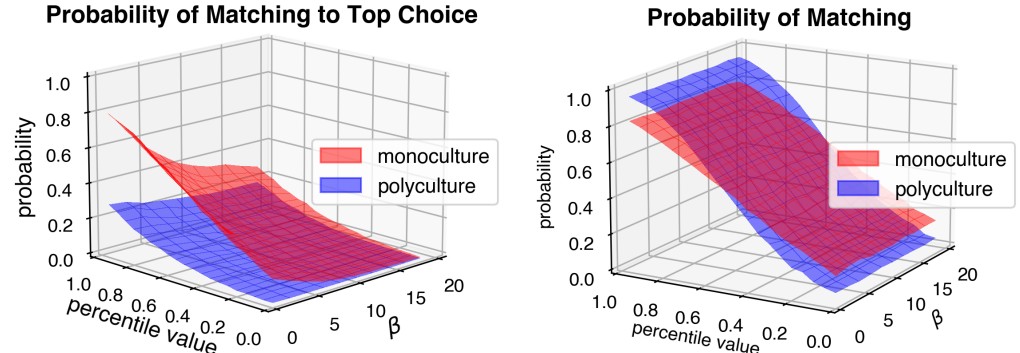

**Figure 8:** The probability an applicant matches to their top choice (right) and to any firm at all (right), as a function of the applicant's percentile true value and $\beta$ the level of global correlation in applicant preferences.

with a firm, what is expected rank of an applicant's match according to their preference list? A lower average rank corresponds to better applicant outcomes, since this means that applicants are on average matched to more-preferred options. As shown in Figure 7 (right), the average rank of applicant matches is worse under polyculture than monoculture for all $\beta$ and $\gamma$ we consider. Notice that for both monoculture and polyculture, the average rank of applicant matches increases as $\beta$ increases. Intuitively, this is true because the increased correlation between applicant preferences means that it is harder for all applicants to receive their preferred options.

We further test more specific predictions from Theorem 2, that an applicant is more likely to be matched to their top choice under monoculture (Theorem 2(i)), and that some applicant's are more likely to be matched overall under polyculture (Theorem 2(iii)), leading to a non-stochastic-dominance result. Our experiments are plotted in Figure 8. We vary the level of global correlation $\beta$ and consider the probability of matching conditional on an applicant's true value. Our experiments confirm the two predictions. Note that as $\beta$ increases, the probability of an applicant matching to their top choice is smaller both for monoculture and polyculture. Since many applicants share the same top choice, few applicants can be matched to their top choice.

### D.3 Differential application access

We now test our third main theoretical prediction: that monoculture is more robust to polyculture with respect to variation in the number of applications submitted by different applicants. Recall that in order to study differential access, it is necessary to specify an application strategy for each applicant—for an applicant that can apply to $k$ firms, which $k$ do they select? In our theoretical setup—where there was no correlation in applicant preferences—we benefited from the existence of a simple Nash equilibrium, where applicants all apply to their top $k$ choices. Given arbitrary correlation structures, the characterization of Nash equilibria remains an open question, and is an area of continued inquiry.[14]

Therefore, we implement two heuristic strategies: applying to one's $k$ most-preferred firms, and applying to $k$ randomly selected firms (in the order of the applicant's true preferences). The former, as we showed in Proposition 6, is a Nash equilibrium when applicant preferences are fully uncorrelated (i.e., when $\beta = \gamma = 0$). Intuitively, this strategy performs poorly when preferences are correlated: applicants who have low value and who can only apply to one firm should not apply to their most preferred firm, which is likely to be highly competitive. In this case, the second heuristic strategy we propose, applying to randomly selected firms (in the true order of the applicant's preferences) is likely to perform better. Roughly, applying randomly implements the commonly-used "reach-match-safety" approach, which has also been theoretically justified by [4].

---

[14][14] and [4] consider the optimal choice of $k$ firms to apply to when admissions probabilities are independent and correlated, respectively. Neither, however, consider interactions between the strategies of multiple applicants. [25] show the existence of Nash equilibria in the deferred acceptance algorithm (in addition to other mechanisms) when applicants can only apply to $k$ firms. However, it does not consider the stochasticity present in our model.

**Applicants Who Submit More Applications Benefit More Under Polyculture**

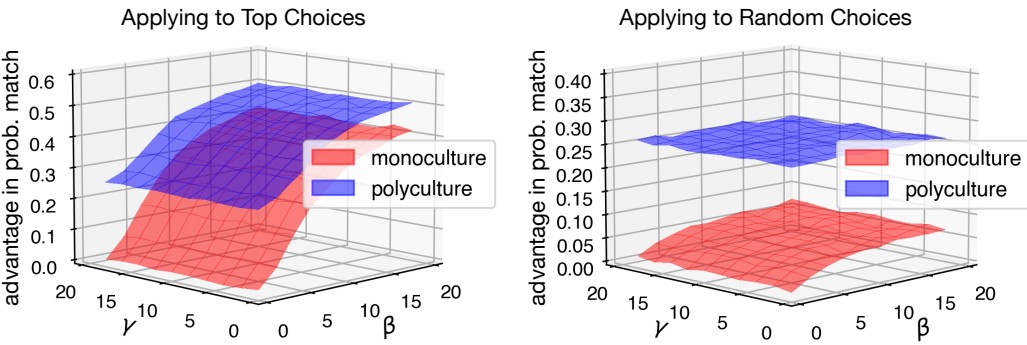

**Figure 9:** Difference in probability of matching among applicants who can apply to between 6 and 10 firms, and those who can apply to between 1 and 5 firms. This difference is always higher in polyculture than in monoculture across the parameters we consider. This holds both when applicants apply to their top choices (left) and when they apply to random firms in their order of preference (right).

**Change in Firm Welfare Under Differential Application Access**

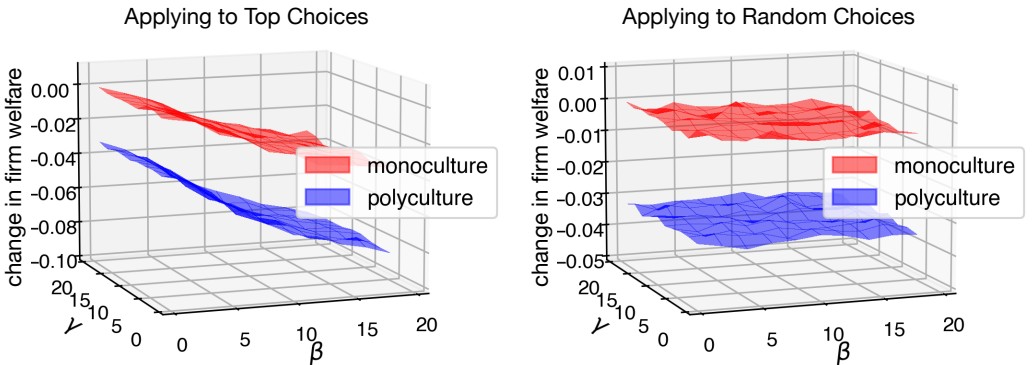

**Figure 10:** The change in college welfare when moving from (1) a market with no differential application access, to (2) a market with differential application access. Note that college welfare decreases more under polyculture. This holds both when applicants apply to their top choices (left) and when they apply to random firms in their order of preference (right).

In our experiments, we consider $\kappa$ to be the uniform distribution on $\{1, 2, \cdots, 10\}$, so applicants are allowed to apply to a random number of firms between 1 and 10. In Figure 9, we plot the difference in the probability of matching between applicants who can apply to between 6 and 10 firms and applicants who can apply to between 1 and 5 firms. This is a measure of the benefit accrued by an applicant who applies to more firms. A larger difference is equivalent to less robustness to differential application access, since this implies that applicants who apply to more firms have a larger chance of being matched. As shown in Figure 9, this difference is larger under polyculture than under monoculture for all choices of $\beta$ and $\gamma$ we consider. This holds both when applicants apply to their most-preferred firms and random firms.

We further test our prediction that firm welfare is affected more by differential application access under polyculture than under monoculture. To do this, we consider the change in firm welfare when moving from "uniform application access" where all applicants apply to all firms to differential application access according to $\kappa$. As shown in Figure 10, this change is more significant under polyculture than under monoculture, for both application strategies. This supports our prediction.

# E  Preliminary Results and Proofs

## E.1  Observations About Measures with Connected Support

To aid subsequent proofs, we recall our assumption on the probability measure $\eta$ (of student values) and the probability distribution $\mathcal{D}$ (the noise distribution); namely, that they have connected support. We then make two useful observations.

Let $\pi$ be the probability measure associated with $\mathcal{D}$. Then we make the assumption that both $\eta$ and $\pi$ have *connected support*, where we mean that the smallest closed set of measure 1 is an interval. Let these respective intervals be $[V_-, V_+]$ and $[X_-, X_+]$. (Note that $V_-, X_-$ may be equal to $-\infty$ and $V_+, X_+$ may be equal to $\infty$.) In what follows we let $V$ and $X$ be random variables distributed according to the probability measures $\eta$ and $\pi$, and let $F_V$ and $F_X$ denote their cdfs.

**Proposition 7.** *The following hold:*

(i) *$F_V$ is strictly increasing on the interval $(V_-, V_+)$, and $F_V(v) \in (0,1)$ when $v \in (V_-, V_+)$.*

(ii) *$F_X$ is strictly increasing on the interval $(X_-, X_+)$, and $F_X(x) \in (0,1)$ when $x \in (X_-, X_+)$.*

(iii) *For all integers $m \geq 1$, $F_X^m$ is strictly increasing on the interval $(X_-, X_+)$, and $F_X^m(x) \in (0,1)$ when $x \in (X_-, X_+)$.*

*Proof.* We show the result for $F_V$; the result for $F_X$ is (clearly) analogous.

First suppose that $F_V$ were not strictly increasing on $V_-, V_+$, such that there exists $v_1, v_2$ such that $V_- < v_1 < v_2 < V_+$ and $F_V(v_1) = F_V(v_2)$. Then $\eta((v_1, v_2)) = F_V(v_2) - F_V(v_1) = 0$. This implies that $\eta([V_-, V_+] \setminus [v_1, v_2]) = 1 - 0 = 1$, which contradicts the assumption that $[V_-, V_+]$ is the smallest closed set of $\eta$-measure 1.

To show that $F_V(v) \in (0,1)$ when $v \in (V_-, V_+)$, it suffices to show that $F_V(V_-) = 0$ and $F_V(V_+) = 1$. Assume otherwise, such that $F_V(V_-) > 0$ or $F_V(V_+) < 1$. This would imply that $\eta([V_-, V_+]) < 1$, which gives a contradiction. $\square$

**Proposition 8.** *Let $I$ be an interval such that*

$$I \cap (V_-, V_+) \neq \emptyset. \tag{17}$$

*Then $\eta(I) > 0$.*

*Proof.* There exists $a, b$ such that $V_- \leq a < b \leq V_+$ such that $(a,b) \subseteq I \cap (V_-, V_+)$. Therefore, $\eta(I) > \eta((a,b))$. We have that $\eta((a,b)) = F_V(b) - F_V(a) > 0$, where we used that $F_V$ is strictly increasing on $[V_-, V_+]$, as shown in Proposition 7. This gives the result. $\square$

## E.2  The Lattice Theorem for Stable Matchings

A powerful result in the stable matching literature shows that stable matchings have a complete lattice structure. In this section, we state this result in our setup (Theorem A.1 in [10]). Consider the lattice operators $\vee$ and $\wedge$ on cutoffs in $\mathbb{R}^C$, where for a set $Z$ of cutoffs,

$$\left( \bigvee_{P \in Z} P \right)_f = \sup_{P \in Z} P_f \tag{18}$$

and

$$\left( \bigwedge_{P \in Z} P \right)_f = \inf_{P \in Z} P_f. \tag{19}$$

**Proposition 9** (The Lattice Theorem). *The set of market-clearing cutoffs forms a complete lattice with respect to $\vee$ and $\wedge$ as defined above.*

For our purposes, this will be useful to establish the two "Equal Cutoffs Lemmas" we prove (Lemma 2 and Lemma 5), in which we show that our economies induce unique stable matchings, in which all cutoffs are equal. The high-level idea is to show that the maximum and minimum cutoff vectors (1) are each comprised of identical cutoffs, and (2) coincide.

# F  Proofs for Section 4

We begin by proving the Equal Cutoffs Lemma for $\theta_{\mathrm{mono}}$ and $\theta_{\mathrm{poly}}$.

***Proof of Lemma 2.*** For now, consider a generic $\theta \in \{\theta_{\mathrm{mono}}, \theta_{\mathrm{poly}}\}$. Let $Z$ be the set of market-clearing cutoffs. Then by symmetry, if $P = (P_1, \cdots, P_m) \in Z$, then any permutation of $P$ is also in $Z$. It follows that

$$\sup_{P \in Z} P_1 = \sup_{P \in Z} P_2 = \cdots = \sup_{P \in Z} P_m \tag{20}$$

and

$$\inf_{P \in Z} P_1 = \inf_{P \in Z} P_2 = \cdots = \inf_{P \in Z} P_m. \tag{21}$$

Call these two common values $P_+$ and $P_-$ respectively. By Proposition 9, $(P_+, \cdots, P_+), (P_-, \cdots, P_-) \in Z$. These are the greatest and least elements of the complete lattice of market-clearing cutoffs. Therefore, it suffices to show that $P_+ = P_-$. We show this separately for $\theta = \theta_{\mathrm{mono}}$ and $\theta = \theta_{\mathrm{poly}}$.

**Case 1: Monoculture.** Consider $\theta = \theta_{\mathrm{mono}}$. It suffices to show that $P_+ = P_-$. Assume for sake of contradiction that $P_+ > P_-$. (By definition, $P_+ \geq P_-$.) By the definition of market clearing, the total measure of students matched is equal to $S$ under either $(P_-, \cdots, P_-)$ or $(P_+, \cdots, P_+)$. Therefore,

$$\int_{\mathbb{R}} \Pr[v + X > P_+] \, d\eta(v) = S = \int_{\mathbb{R}} \Pr[v + X > P_-] \, d\eta(v). \tag{22}$$

To give the desired contradiction, we show that

$$\int_{\mathbb{R}} \Pr[v + X > P_+] \, d\eta(v) < \int_{\mathbb{R}} \Pr[v + X > P_-] \, d\eta(v). \tag{23}$$

Clearly, $\Pr[v + X > P_+] \leq \Pr[v + X > P_-]$. Therefore, it suffices to show that $\Pr[v + X > P_+] < \Pr[v + X > P_-]$ on a set $I$ of positive $\eta$-measure. We show this is the case for $I = (P_- - X_+, P_- - X_-)$.

We first show that the inequality is satisfied on $I$. Note that for $v \in I$, we have that $P_- - v \in (X_-, X_+)$. Therefore,

$$\Pr[v + X > P_-] - \Pr[v + X > P_+] = F_X(P_+ - v) - F_X(P_- - v) > 0 \tag{24}$$

where the last inequality follows from observing that by $P_+ - v > P_- - v$ by assumption and applying Proposition 7, which says that $F_X$ is strictly increasing on $(X_-, X_+)$.

It remains to show that the interval $I$ has positive $\eta$-measure. We begin by showing that $P_- > V_- + X_-$. This is true because for $P \leq V_- + X_-$,

$$\int_{\mathbb{R}} \Pr[v + X > P] \, d\eta(v) \geq \int_{\mathbb{R}} \Pr[v + X > V_- + X_-] \, d\eta(v) = \int_{\mathbb{R}} 1 \, d\eta(v) = 1, \tag{25}$$

contradicting the assumption that $S < 1$. Similarly, $P_- < V_+ + X_+$. It follows that $P_- - X_+ < V_+$ and $P_- - X_- > V_-$. Therefore, the interval $I = (P_- - X_+, P_- - X_-)$ intersects $(V_-, V_+)$. So Proposition 8 implies that it has positive measure.

**Case 2: Polyculture.** The result holds analogously for $\theta = \theta_{\mathrm{poly}}$, where we replace $X$ with $X^{(m)}$ (and thus $F_X$ with $F_X^m$).

It suffices to show that $P_+ = P_-$. Assume for sake of contradiction that $P_+ > P_-$. (By definition, $P_+ \geq P_-$.) By the definition of market clearing, the total measure of students matched is equal to $S$ under either $(P_-, \cdots, P_-)$ or $(P_+, \cdots, P_+)$. Therefore,

$$\int_{\mathbb{R}} \Pr[v + X^{(m)} > P_+] \, d\eta(v) = S = \int_{\mathbb{R}} \Pr[v + X^{(m)} > P_-] \, d\eta(v). \tag{26}$$

To give the desired contradiction, we show that

$$\int_{\mathbb{R}} \Pr[v + X^{(m)} > P_+] \, d\eta(v) < \int_{\mathbb{R}} \Pr[v + X^{(m)} > P_-] \, d\eta(v). \tag{27}$$

Clearly, $\Pr[v + X^{(m)} > P_+] \leq \Pr[v + X^{(m)} > P_-]$. Therefore, it suffices to show that $\Pr[v + X^{(m)} > P_+] < \Pr[v + X^{(m)} > P_-]$ on a set $I$ of positive $\eta$-measure. We show this is the case for $I = (P_- - X_+, P_- - X_-)$.

We first show that the inequality is satisfied on $I$. Note that for $v \in I$, we have that $P_- - v \in (X_-, X_+)$. Therefore,

$$\Pr[v + X^{(m)} > P_-] - \Pr[v + X^{(m)} > P_+] = F_X^m(P_+ - v) - F_X^m(P_- - v) > 0 \quad (28)$$

where the last inequality follows from observing that by $P_+ - v > P_- - v$ by assumption and applying Proposition 7, which says that $F_X^m$ is strictly increasing on $(X_-, X_+)$.

It remains to show that the interval $I$ has positive $\eta$-measure. We begin by showing that $P_- > V_- + X_-$. This is true because for $P \leq V_- + X_-$,

$$\int_{\mathbb{R}} \Pr[v + X^{(m)} > P] \, d\eta(v) \geq \int_{\mathbb{R}} \Pr[v + X^{(m)} > V_- + X_-] \, d\eta(v) = \int_{\mathbb{R}} 1 \, d\eta(v) = 1, \quad (29)$$

contradicting the assumption that $S < 1$. Similarly, $P_- < V_+ + X_+$. It follows that $P_- - X_+ < V_+$ and $P_- - X_- > V_-$. Therefore, the interval $I = (P_- - X_+, P_- - X_-)$ intersects $(V_-, V_+)$. So Proposition 8 implies that it has positive measure. $\qquad\square$

As a consequence of the above proof,

$$\eta((P_{\mathrm{mono}} - X_+, P_{\mathrm{mono}} - X_-)) > 0 \quad (30)$$

and

$$\eta((P_{\mathrm{poly}} - X_+, P_{\mathrm{poly}} - X_-)) > 0, \quad (31)$$

which will be useful in subsequent proofs.

***Proof of Corollary 4.*** Noting that the total measure of students matched is the same under both monoculture and polyculture (equal to $S$), we have, using (5) and (6) that

$$\int_{\mathbb{R}} \Pr[v + X > P_{\mathrm{mono}}] \, d\eta(v) = S = \int_{\mathbb{R}} \Pr[v + X^{(m)} > P_{\mathrm{poly}}] \, d\eta(v). \quad (32)$$

Assume for the sake of contradiction that $P_{\mathrm{mono}} \geq P_{\mathrm{poly}}$. To give the desired contradiction, we will show that this implies

$$\int_{\mathbb{R}} \Pr[v + X > P_{\mathrm{mono}}] \, d\eta(v) < \int_{\mathbb{R}} \Pr[v + X^{(m)} > P_{\mathrm{poly}}] \, d\eta(v). \quad (33)$$

Clearly,

$$\Pr[v + X > P_{\mathrm{mono}}] \leq \Pr[v + X^{(m)} > P_{\mathrm{poly}}] \quad (34)$$

for all $v$. We now show that

$$\Pr[v + X > P_{\mathrm{mono}}] < \Pr[v + X^{(m)} > P_{\mathrm{poly}}] \quad (35)$$

for $v \in (P_{\mathrm{poly}} - X_+, P_{\mathrm{poly}} - X_-)$, an interval which we will show has positive $\eta$-measure. We have that

$$\Pr[v + X > P_{\mathrm{mono}}] \leq \Pr[v + X > P_{\mathrm{poly}}], \quad (36)$$

so it suffices to show that

$$\Pr[v + X > P_{\mathrm{poly}}] < \Pr[v + X^{(m)} > P_{\mathrm{poly}}]. \quad (37)$$

We have that

$$\Pr[v + X > P_{\mathrm{poly}}] = 1 - F_X(P_{\mathrm{poly}} - v) \quad (38)$$

$$\Pr[v + X^{(m)} > P_{\mathrm{poly}}] = 1 - F_X^m(P_{\mathrm{poly}} - v). \quad (39)$$

Therefore, since $m \geq 2$, (37) holds whenever $F_X(P_{\mathrm{poly}} - v) \in (0, 1)$, which, by Proposition 7, is true when $P_{\mathrm{poly}} - v \in (X_-, X_+)$. Equivalently, $v \in (P_{\mathrm{poly}} - X_+, P_{\mathrm{poly}} - X_-)$. This interval has positive measure (31). $\qquad\square$

### F.1 Proof of Theorem 1

We first prove part (i) and then part (ii).

#### F.1.1 Theorem 1(i)

The proof of Theorem 1(i) requires a few key observations, which we provide as separate (but closely related) parts in the following lemma.

**Lemma 10.** *If $\mathcal{D}$ is maximum concentrating, the following hold for all $\varepsilon, \delta > 0$ when $m$ is sufficiently large:*

*(i) For all $v < P_{\text{poly}} - \mathbb{E}\left[X^{(m)}\right] - \frac{\delta}{3}$,*

$$\Pr[\mu_{\text{poly}}(v) \in m] < \varepsilon. \tag{40}$$

*(ii) For all $v > P_{\text{poly}} - \mathbb{E}\left[X^{(m)}\right] + \frac{\delta}{3}$,*

$$\Pr[\mu_{\text{poly}}(v) \in m] > 1 - \varepsilon. \tag{41}$$

*(iii)*

$$\left|\left(P_{\text{poly}} - \mathbb{E}\left[X^{(m)}\right]\right) - v_S\right| < \frac{2\delta}{3}. \tag{42}$$

We provide some intuition for each part of the lemma, and how they come together to show Theorem 1. First consider the value $P_{\text{poly}} - \mathbb{E}\left[X^{(m)}\right]$, which is a natural "threshold" in the following sense: The expected maximum score of a student with value $P_{\text{poly}} - \mathbb{E}\left[X^{(m)}\right]$ is exactly the cutoff $P_{\text{poly}}$. Part (i) of the lemma says that a student with even a slightly lower score than $P_{\text{poly}} - \mathbb{E}\left[X^{(m)}\right]$ has a very low chance of having their maximum score exceed the cutoff $P_{\text{poly}}$. Meanwhile, part (ii) says that a student with even a slightly higher score than $P_{\text{poly}} - \mathbb{E}\left[X^{(m)}\right]$ is almost certain to have their maximum score exceed $P_{\text{poly}}$. Part (iii) of the lemma says that this threshold $P_{\text{poly}} - \mathbb{E}\left[X^{(m)}\right]$ cannot differ much from $v_S$, which we recall is defined such that exactly an $S$ proportion of students have value higher than $v_S$. Parts (i) and (ii) essentially show that there is a threshold at which the probability a student is matched jumps from nearly 0 to almost 1; part (iii) specifies the location of this threshold. In combination, these parts show Theorem 1:

***Proof of Theorem 1 using Lemma 10.*** We first show that for $v < v_S$, for all $\varepsilon > 0$, $\Pr[\mu_{\text{poly}}(v) \in \mathcal{F}] < \varepsilon$ for all $m$ sufficiently large. Consider any $v < v_S$. Then there exists $\delta > 0$ such that $v < v_S - \delta$. Using Lemma 10(iii) and taking $m$ sufficiently large, it follows from $v < v_S - \delta$ that $v < P_{\text{poly}} - \mathbb{E}\left[X^{(m)}\right] - \frac{\delta}{3}$. In turn, again taking $m$ sufficiently large, Lemma 10(i) implies that

$$\Pr[\mu_{\text{poly}}(v) \in \mathcal{F}] < \varepsilon. \tag{43}$$

We next show that for $v > v_S$, for all $\varepsilon > 0$, $\Pr[\mu_{\text{poly}}(v) \in \mathcal{F}] > 1 - \varepsilon$ for all $m$ sufficiently large. Consider any $v > v_S$. Then there exists $\delta > 0$ such that $v > v_S + \delta$. Taking $m$ sufficiently large, Lemma 10(iii) shows that $v > v_S + \delta$ implies that $v > P_{\text{poly}} - \mathbb{E}\left[X^{(m)}\right] + \frac{\delta}{3}$, and Lemma 10(ii) implies that

$$\Pr[\mu_{\text{poly}}(v) \in \mathcal{F}] > 1 - \varepsilon. \tag{44}$$

This completes the proof. □

***Proof of Lemma 10.*** We show the three parts in order.

We first show part (i). Suppose that $v < P_{\text{poly}} - \mathbb{E}\left[X^{(m)}\right] - \frac{\delta}{3}$. Then, using Proposition 3(ii),

$$\Pr\left[\mu_{\text{poly}}(v) \in \mathcal{F}\right] = \Pr\left[v + \max_{f \in \mathcal{F}} X_f > P_{\text{poly}}\right] \tag{45}$$

$$= \Pr\left[X^{(m)} > P_{\text{poly}} - v\right] \tag{46}$$

$$< \Pr\left[X^{(m)} > \mathbb{E}\left[X^{(m)}\right] + \frac{\delta}{3}\right] \tag{47}$$

$$= \Pr\left[X^{(m)} - \mathbb{E}\left[X^{(m)}\right] > \frac{\delta}{3}\right] \tag{48}$$

$$< \varepsilon, \tag{49}$$

where the last line follows from the definition of $\mathcal{D}$ being maximum concentrating and by taking $m$ sufficiently large.

Part (ii) can be shown analogously.

To show part (iii), we use parts (i) and (ii). For $m$ sufficiently large,

$$S = \int_{-\infty}^{\infty} \Pr[\mu_{\text{poly}}(v) \in \mathcal{F}]\, d\eta(v) \tag{50}$$

$$= \int_{-\infty}^{P_{\text{poly}} - \mathbb{E}\left[X^{(m)}\right] - \frac{\delta}{3}} \Pr[\mu_{\text{poly}}(v) \in \mathcal{F}]\, d\eta(v) + \int_{P_{\text{poly}} - \mathbb{E}\left[X^{(m)}\right] - \frac{\delta}{3}}^{\infty} \Pr[\mu_{\text{poly}}(v) \in \mathcal{F}]\, d\eta(v) \tag{51}$$

$$< \int_{-\infty}^{P_{\text{poly}} - \mathbb{E}\left[X^{(m)}\right] - \frac{\delta}{3}} \varepsilon\, d\eta(v) + \int_{P_{\text{poly}} - \mathbb{E}\left[X^{(m)}\right] - \frac{\delta}{3}}^{\infty} 1\, d\eta(v) \tag{52}$$

$$= \eta\left(\left(-\infty, P_{\text{poly}} - \mathbb{E}\left[X^{(m)}\right] - \frac{\delta}{3}\right)\right) \cdot \varepsilon + \eta\left(\left(P_{\text{poly}} - \mathbb{E}\left[X^{(m)}\right] - \frac{\delta}{3}, \infty\right)\right), \tag{53}$$

where (52) follows from part (i) and by taking $m$ sufficiently large.

Now define $A := \eta\left(\left(P_{\text{poly}} - \mathbb{E}\left[X^{(m)}\right] - \frac{\delta}{3}, \infty\right)\right)$. Substituting this notation into (53), we have

$$S < (1 - A) \cdot \varepsilon + A \tag{54}$$

$$S - \varepsilon < (1 - \varepsilon)A \tag{55}$$

$$A > \frac{S - \varepsilon}{1 - \varepsilon}. \tag{56}$$

Taking $\varepsilon$ sufficiently small, $A$ is arbitrarily close to $S$, which implies—using that $\eta$ has connected support—that $P_{\text{poly}} - \mathbb{E}\left[X^{(m)}\right] - \frac{\delta}{3} < v_S + \delta'$ for any $\delta' > 0$ when $m$ is sufficiently large. Setting $\delta' = \frac{\delta}{3}$, we have that

$$\left(P_{\text{poly}} - \mathbb{E}\left[X^{(m)}\right]\right) - v_S < \frac{2\delta}{3}. \tag{57}$$

when $m$ is sufficiently large.

We may analogously use (ii) to show that

$$v_S - \left(P_{\text{poly}} - \mathbb{E}\left[X^{(m)}\right]\right) < \frac{2\delta}{3}. \tag{58}$$

Combining (57) and (58) shows the result. $\qquad\square$

### F.1.2 Theorem 1(ii)

*Proof of Theorem 1(ii).* We first show that $P_{\text{mono}}$ is fixed for all $m$. Indeed,

$$S = \int_{-\infty}^{\infty} \Pr[\mu_{\text{mono}}(v) \in \mathcal{F}]\, d\eta(v) \tag{59}$$

$$= \int_{-\infty}^{\infty} \Pr[v + X > P_{\text{mono}}]\, d\eta(v). \tag{60}$$

It follows from the proof of Lemma 2 that there is a unique solution to (60). Therefore, $P_{\text{mono}}$ is the same for all $m$. It follows directly that $\Pr[v + X > P_{\text{mono}}]$ is constant in $m$.

It remains to show that college welfare is suboptimal for $\mu_{\text{mono}}$—i.e., that

$$\int_{-\infty}^{\infty} v \Pr[v + X > P_{\text{mono}}] \, d\eta(v) < \int_{v_S}^{\infty} v \, d\eta(v). \tag{61}$$

Since $\int_{-\infty}^{\infty} \Pr[v + X > P_{\text{mono}}] \, d\eta(v) = S = \int_{v_S}^{\infty} v \, d\eta(v)$, it suffices to show that $\Pr[v + X > P_{\text{mono}}] \in (0,1)$ on a set $I$ of positive $\eta$-measure. The inequality clearly holds for $I = (P_{\text{mono}} - X_+, P_{\text{mono}} - X_-)$, where we use that $X$ has connected support $[X_-, X_+]$. (Specifically, Proposition 7 gives that $F_X(x)$ is strictly increasing and lies in $(0,1)$ for $x \in (X_-, X_+)$.) Finally, we have from (30) that $I$ has positive measure. $\qquad\square$

### F.2 Proofs for Section 4.2

*Proof of Theorem 2.* We show the three parts separately:

**Part (i).** To show part (i), note that

$$\Pr[\text{Rank}_v(\mu_{\text{mono}}(v)) = 1] = \Pr[v + X > P_{\text{mono}}]. \tag{62}$$

This can be contrasted with the polyculture setting, in which

$$\Pr[\text{Rank}_v(\mu_{\text{poly}}(v)) = 1] = \Pr[v + X > P_{\text{poly}}]. \tag{63}$$

By Corollary 4, $P_{\text{mono}} < P_{\text{poly}}$ for all $m$, from which the non-strict inequality. To characterize when the inequality is strict, note that

$$\Pr[\text{Rank}_v(\mu_{\text{mono}}(v)) = 1] = \Pr[v + X > P_{\text{mono}}] = 1 - F_X(P_{\text{mono}} - v) \tag{64}$$

$$\Pr[\text{Rank}_v(\mu_{\text{poly}}(v)) = 1] = \Pr[v + X > P_{\text{poly}}] = 1 - F_X(P_{\text{poly}} - v). \tag{65}$$

When $v \in (P_{\text{mono}} - X_+, P_{\text{mono}} - X_-)$, we have that

$$P_{\text{mono}} - v \in (X_-, X_+). \tag{66}$$

Noting that $P_{\text{poly}} - v > P_{\text{mono}} - v$, the result follows from Proposition 7, which shows that $F_X$ is strictly increasing on $(X_-, X_+)$. Also, we have already shown that this interval is of positive measure (30).

**Part (ii).** Part (ii) is a direct consequence of the Equal Cutoffs Lemma (Lemma 2), since if, under monoculture, the student's score is above the cutoff for one college, then it is above the cutoff for all colleges. Therefore, any student who receives an offer from one college receives an offer from all colleges, and matches with their top choice.

**Part (iii).** We first show that $v_S < P_{\text{mono}} - X_-$. Indeed,

$$\int_{P_{\text{mono}} - X_-}^{\infty} 1 \, d\eta(v) = \int_{P_{\text{mono}} - X_-}^{\infty} \Pr[\mu_{\text{mono}}(v) \in \mathcal{F}] \, d\eta(v) \tag{67}$$

$$< S \tag{68}$$

$$= \int_{v_S}^{\infty} 1 \, d\eta(v). \tag{69}$$

It is also clear that $v_S < V_+$. Therefore, $(v_S, P_{\text{mono}} - X_-)$ is a non-empty interval that intersects $(V_-, V_+)$, and so has positive $\eta$-measure from Proposition 8. It suffices to show that $\Pr[\mu_{\text{mono}}(v) \in \mathcal{F}] < \Pr[\mu_{\text{poly}}(v) \in \mathcal{F}]$ on this interval.

When $v < P_{\text{mono}} - X_-$,

$$\Pr[\mu_{\text{mono}}(v) \in \mathcal{F}] = 1 - F_X(P_{\text{mono}} - v) < 1. \tag{70}$$

Note here that $1 - F_X(P_{\text{mono}} - v)$ is constant in $m$, so for all $v \in (v_S, P_{\text{mono}} - X_-)$,

$$\lim_{C \to \infty} \Pr[\mu_{\text{mono}}(v) \in \mathcal{F}] < 1. \tag{71}$$

Meanwhile, for any $v > v_S$, Theorem 1 gives that

$$\lim_{C \to \infty} \Pr[\mu_{\text{poly}}(v) \in \mathcal{F}] = 1. \tag{72}$$

It follows that for all $(v_S, P_{\text{mono}} - X_-)$, for $m$ sufficiently large,

$$\Pr[\mu_{\text{poly}}(v) \in \mathcal{F}] > \Pr[\mu_{\text{mono}}(v) \in \mathcal{F}]. \tag{73}$$

$\square$

## G  Proofs for Section A

***Proof of Lemma 5.*** Let $Z$ be the set of market-clearing cutoffs. Then by symmetry, if $P = (P_1, \cdots, P_m) \in Z$, then any permutation of $P$ is also in $Z$. It follows that

$$\sup_{P \in Z} P_1 = \sup_{P \in Z} P_2 = \cdots = \sup_{P \in Z} P_m \tag{74}$$

and

$$\inf_{P \in Z} P_1 = \inf_{P \in Z} P_2 = \cdots = \inf_{P \in Z} P_m. \tag{75}$$

Call these two common values $P_+$ and $P_-$ respectively. By Proposition 9, $(P_+, \cdots, P_+), (P_-, \cdots, P_-) \in Z$. These are the greatest and least elements of the complete lattice of market-clearing cutoffs. Therefore, it suffices to show that $P_+ = P_-$.

We show the result for $\theta = \theta_{\text{poly}, \kappa}$, and the result follows similarly for $\theta = \theta_{\text{mono}, \kappa}$. It suffices to show that $P_+ = P_-$. Assume for sake of contradiction that $P_+ > P_-$. (By definition, $P_+ \geq P_-$.) By the definition of market clearing, the total measure of students matched is equal to $S$ under either $(P_-, \cdots, P_-)$ or $(P_+, \cdots, P_+)$. Therefore,

$$\int_{\mathbb{R}} \Pr[v + X^{(\kappa)} > P_+] \, d\eta(v) = S = \int_{\mathbb{R}} \Pr[v + X^{(\kappa)} > P_-] \, d\eta(v). \tag{76}$$

To give the desired contradiction, we show that

$$\int_{\mathbb{R}} \Pr[v + X^{(\kappa)} > P_+] \, d\eta(v) < \int_{\mathbb{R}} \Pr[v + X^{(\kappa)} > P_-] \, d\eta(v). \tag{77}$$

Clearly, $\Pr[v + X^{(\kappa)} > P_+] \leq \Pr[v + X^{(\kappa)} > P_-]$. Therefore, it suffices to show that $\Pr[v + X^{(\kappa)} > P_+] < \Pr[v + X^{(\kappa)} > P_-]$ on a set $I$ of positive $\eta$-measure. We show this is the case for $I = (P_- - X_+, P_- - X_-)$.

We first show that the inequality is satisfied on $I$. Note that for $v \in I$, we have that $P_- - v \in (X_-, X_+)$. Therefore,

$$\Pr[v + X^{(\kappa)} > P_-] - \Pr[v + X^{(\kappa)} > P_+] = \sum_{k=1}^{m} \Pr[\kappa = k] \left( F_X^k(P_+ - v) - F_X^k(P_- - v) \right) > 0 \tag{78}$$

where the last inequality follows from observing that by $P_+ - v > P_- - v$ by assumption and applying Proposition 7, which says that $F_X^k$ is strictly increasing on $(X_-, X_+)$.

It remains to show that the interval $I$ has positive $\eta$-measure. We begin by showing that $P_- > V_- + X_-$. This is true because for $P \leq V_- + X_-$,

$$\int_{\mathbb{R}} \Pr[v + X^{(\kappa)} > P] \, d\eta(v) \geq \int_{\mathbb{R}} \Pr[v + X > V_- + X_-] \, d\eta(v) = \int_{\mathbb{R}} 1 \, d\eta(v) = 1, \tag{79}$$

contradicting the assumption that $S < 1$. Similarly, $P_- < V_+ + X_+$. It follows that $P_- - X_+ < V_+$ and $P_- - X_- > V_-$. Therefore, the interval $I = (P_- - X_+, P_- - X_-)$ intersects $(V_-, V_+)$. So Proposition 8 implies that it has positive measure. $\square$

***Proof of Theorem 3.*** To show part (i), note that $P_{\text{mono}} = P_{\text{mono}, \kappa}$.

To show part (ii), note that

$$\Pr[\mu_{\mathrm{poly},\kappa}(v) \in \mathcal{F} \mid k(v) = k] = \Pr[v + X^{(k)} > P_{\mathrm{poly},\kappa}] \tag{80}$$

$$= 1 - F_X^k(P_{\mathrm{poly},\kappa} - v), \tag{81}$$

which is increasing in $k$. Moreover, when $v \in (P_{\mathrm{poly},\kappa} - X_+, P_{\mathrm{poly},\kappa} - X_-)$, we have that $P_{\mathrm{poly},\kappa} - v \in (X_-, X_+)$, so by Proposition 7, $F_X(P_{\mathrm{poly},\kappa}) \in (0, 1)$, in which case (81) is strictly increasing in $k$. $\qquad\square$

