# OpenReview forum: "Monoculture in Matching Markets"
_NeurIPS.cc/2024/Conference — NeurIPS 2024 poster_

### Official Review · Reviewer_LFB2 · 2024-06-23

**Soundness:** 4
**Presentation:** 4
**Contribution:** 4
**Rating:** 7
**Confidence:** 4

**Summary:**

The authors are focused on matching markets in which different firms use a single algorithm / evaluation criterion (monoculture) vs. markets where different firms may each have different evaluation algorithms / criterion (polyculture). This can be seen as a substantial generalization of the wonderful work of Kleinberg and Raghavan [35] on monoculture in hiring with two firms.

The authors first introduce the continuum matching market model introduced by Azevedo and Leshno [10]. Here, there is a continuum of applicants, and a finite number of firms. The authors make the assumptions that (1) each firm has the same fixed capacity, and (2) not all applicants will eventually be matched with a firm. The authors first review critical results in the existing continuum model. These include the fact that a stable matching corresponds to a particular cutoff vector, and subject to this cutoff vector, applicants always choose their highest preferred firm for which their estimated quality is higher than the cutoff for that firm.

Next, the authors introduce their notion of mono and polyculture into this model. Intuitively, monoculture is where each firm has an identical estimate of the value of an applicant of type $\theta(v)$, given by $v + X$ for an $X$ drawn from some noise distribution $D$. This captures, for example, each firm using Chat GPT to evaluate the resumes of all applicants. In polyculture, each firm $i$ may have a different estimate $v + X_i$ for the value of applicants of type $\theta(v)$.

The authors begin by proving that the cutoff characterization of stable matching is unique in the mono and polyculture settings (Lemma 2). This follows from the lattice structure of stable matchings. Then, in Proposition 3, they show that the probability of an applicant of type $\theta(v)$ being matched under polyculture is related to the maximum of $X_i$ over all firms' noisy estimates $X_i$, whereas under monoculture this probability is related only to $X$ (since all firms have identical estimates).

We now move to the main results. In Theorem 1, the authors show that under polyculture, as the number of firms $m \to \infty$, the (firm-) optimal welfare can be achieved by the resulting matching. This does not hold for monoculture. In particular, under monoculture, the probability that an individual of type $\theta(v)$ is matched at all is constant for varying $m$.
In Theorem 2, the authors examine applicant welfare. They show that applicants have a higher chance of being matched with their top choice under monoculture, but that for a subset of applicants of positive measure, the variance in whether they are matched or not is higher under monoculture than polyculture. This means that not all applicants are incentivized to prefer monoculture unconditionally.

Finally, some extensions under a differential application access setting are provided. Intuitively, the authors show that more applications do not help applicants under monoculture but does under polyculture. Experiments complement most of the theoretical results, and also demonstrate that the uniform preference assumption is not essential to practical relevance of the results.

**Strengths:**

The paper is generally extremely well written, motivated, and clear. I also think that the work is already very important in the modern context in which universities and hiring managers may already be using one of only a handful of services to conduct automated applicant filtering. The work examines what this would potentially lead to in terms of macroeconomic market dynamics.

The theoretical results are presented clearly, and I understood most even though I have not personally worked in the continuum matching model (have only worked in the discrete matching model). I appreciate that the authors also empirically investigate the (strong) assumption that all applicants' preferences over firms are drawn uniformly at random. The empirical results confirm that this is perhaps not a fundamental assumption, even though the (current) proofs critically hinge on it.

This work certainly challenged my preconceived notion (“monoculture=bad”) in a fundamental way and may open a more general line of inquiry into monoculture more broadly. This paper was a pleasure to read, and I look forward to additional work from the authors.

**Weaknesses:**

Note that I did not carefully check the proofs.

(W1) I think the introduction of the continuum model could be made a bit more clear, in particular the definition of applicant types. For example, there seems to be a small typo in lines 141-142:: “The realization of θ(v) is their type, which lies in $\Theta \coloneqq \mathcal{R} \times \mathbb{R}^m$ is the set of applicant types,”.

Further, I am not sure if this is a typo as well (in line 143): “$\succ^\theta$ is the preference ordering of v over firms…”. Do all applicants of value $v$ have the same type $\theta(v)$? That is, do all have the same preferences over firms? Or, do we draw different preferences uniformly at random for each individual of value $v$? These were not clear from just this introduction on the continuum model.

(W2) The model considers identical noise across all applicant “types”. This is certainly a reasonable form of polyculture to analyze, however, in practice we may be more concerned with bias based on different “types” of individuals. I.e., historically underrepresented minorities having a skewed or higher variance noise distribution. “Types” in this sense is (I believe) not captured by solely preference and firm quality estimates. This is certainly less of a weakness and more of a direction of future work, but I think it is perhaps important to mention. The authors have some discussion in lines 350-352, but more could certainly be added earlier in the paper.

(W3) I think that the assumptions made throughout the work are sprinkled throughout the paper. Having a collection of all assumptions, perhaps in the appendix, may help the reader better understand the limitations of the work.

Minor: Most non-theorems from the main paper are referred to incorrectly in the appendix, e.g. Lemma 2 is mistakenly referred to as proposition 2 in the appendix, and similarly proposition 10 / lemma 10, and corollary 4 / proposition 4.

**Questions:**

Q1: How does Lemma 2 (Equal Cutoffs Lemma) relate to Theorem 1 part 1 from Azevedo and Leshno [10], which says that if $\eta$ has full support, then there is a unique stable matching?

Q2: Do we expect the maximum concentrating distribution assumption to hold in, e.g., the included experiments?

**Limitations:**

I do not believe that the authors have a dedicated limitations section of their paper, which I recommend. Some limitations are sprinkled throughout the paper (e.g. Line 350-353), but perhaps more could be mentioned. In particular, how strong/important the technical assumptions are in practice are could be explicitly discussed in a formal limitation section.

---

> ### Author Rebuttal · Authors · 2024-08-07
>
> We appreciate the thoughtful feedback, and are glad you enjoyed the paper. We answer your questions below:
>
> **Q1: How does Lemma 2 (Equal Cutoffs Lemma) relate to Theorem 1 part 1 from Azevedo and Leshno [10], which says that if has full support, then there is a unique stable matching?**
>
> It is essentially implied (since the existence of unequal cutoffs would then imply multiple stable matchings). We can make note of this. We include a standalone proof, partly because it gives some intuition on how to analyze the noisy model we have, and partly because our proof provides some additional facts (equations 30 and 31) that are important for later results.
>
> **Q2: Do we expect the maximum concentrating distribution assumption to hold in, e.g., the included experiments?**
>
> We think that in many cases, we might intuitively expect the maximum order statistic to at least partially concentrate (i.e., have diminishing variance); this is the case when the “maximum evaluation” of an applicant is less noisy than a single evaluation, which seems generally plausible. If we think noise is Gaussian, as is often assumed (though perhaps primarily for tractability), then it would fully concentrate. Theoretically, max-concentration does not hold for noise distributions where tails are “heavier than exponential”, such as the Laplace distribution.
>
> Our ML experiments, in which we did not hard-code any notion of noise (in fact, noise is not even i.i.d.), seem to suggest maximum-concentrating behavior holds in that setup, since the market as a whole tends to select higher-value applicants than individual ML models. It would be interesting to better understand what noise looks like in practice.
>
>
> **Other comments.** Thank you for pointing out potential areas of confusion in our model, which we will address. Applicants of the same true value do not have the same estimated value. What we care about is the distribution of outcomes for a student with true value v. We will make this more clear in the text.
>
> Thank you also for the suggestion about a limitations section. We will add in a limitations section that discusses the following:
> - The technical assumptions (e.g., homogeneous firms, max-concentrating noise), and when they are more/less likely to hold.
> - A summary of what our computational experiments address (varying correlation structure in applicant preferences, estimates derived from ML models).
> - Broader limitations (e.g., heterogeneous noise/bias across applicants, notions of welfare beyond utilitarian frameworks, potential degradation of effects due to search frictions).

---

> > ### Comment · Reviewer_LFB2 · 2024-08-07
> >
> > I thank the authors for the careful response. In general, I agree with the sentiment that simple proofs are in some sense desirable. Thank you for also explaining the relation (and differences) to Castera et al. I am happy to keep my score as is.

---

> > > ### Author Response · Authors · 2024-08-08
> > >
> > > We appreciate the reply, and are glad that you found our responses helpful. Thanks again for the comments.

---

### Official Review · Reviewer_FF9T · 2024-07-11

**Soundness:** 3
**Presentation:** 3
**Contribution:** 3
**Rating:** 5
**Confidence:** 3

**Summary:**

This paper studies the monoculture problem in matching markets from a theoretical perspective. The authors found that on the firms' (colleges') side, monoculture may decrease the quantity of matched applicants; while on the applicants' side, monoculture may help matched applicants to match with higher-ranked firms (colleges). Additionally, monoculture may decrease the risk of unfairness if some applicants are born with more opportunities to apply to more firms (colleges).

**Strengths:**

1. The paper studies the monoculture problem with multiple (possibly infinite) homogeneous firms and heterogeneous applicants differed by real-numbered type, broadening the elements modeling within the monoculture literature.
2. The paper provides positive theoretical and empirical results for monoculture, which are inspiring as the results are counter-intuitive and challenged the preexisting beliefs about monoculture.

**Weaknesses:**

**General Weakness:**

1. The connection between this work and CS/ML conferences is vague, as this paper mainly addresses monoculture, which is intrinsically an economic problem. Additionally, the technical derivation seems quite straightforward.
2. In the model, firms are assumed to be homogeneous, which might be an oversimplification and not realistic.

**Corrections for Typos:**

1. In line 203, it should state "the cutoff under monoculture must be lower."

**Questions:**

See above Weakness.

**Limitations:**

One concern is whether the paper is a good fit for NeurIPS acceptance. The paper focuses more on economics and matching markets than on machine learning and computer science, attempting to analyze the phenomenon of monoculture using economic models. This paper might be more suitable for conferences like EC and WINE.

---

> ### Author Rebuttal · Authors · 2024-08-07
>
> We appreciate the reviewers comments, and are glad to hear that it challenged preexisting beliefs about monoculture. We address your comments below:
>
> **Assumptions/Straightforwardness:** We note in our general response that we believe our computational experiments demonstrate that our conceptual insights extend beyond our theoretical assumptions. We also noted in the general response that we believe the technical straightforwardness is only possible due to key insights.
>
> **Fit for NeurIPS:** Regarding the fit for NeurIPS, we note that papers studying monoculture have appeared in several ML venues. For example, Bommasani et al. (NeurIPS 2022), Jagadeesan et al. (NeurIPS 2023), Toups et al. (NeurIPS 2023) Jain et al. (ICML 2024), and Jain et al. (Best Paper at FAccT 2024). Insofar as the ML community is interested in questions related to algorithmic hiring and monoculture, we believe that the market-level approach we present is essential for understanding effects. Indeed, while this prior work exclusively views monoculture as bad for applicants, our work (as you note) demonstrates that market-level effects may reverse (and add nuance to) this effect.
>
> In fact, we directly seek to connect to this literature in our paper by adapting the setup of Bommasani et al. into our framework in our computational ML-based experiments.
>
> More broadly, this paper fits into the “social and economic aspects of machine learning” and “theory” parts of the Neurips CFP, the latter of which further specifies Algorithmic Game Theory (which arguably is even more an “EC” topic than our paper). We also note that NeurIPS accepts matching papers that intersect with ML, e.g.: Learning equilibria in matching markets from bandit feedback. M Jagadeesan, A Wei, Y Wang, M Jordan, J Steinhardt (NeurIPS 2021).
>
> (Thanks also for pointing out the typo, which we will correct.)

---

> > ### Comment · Reviewer_FF9T · 2024-08-08
> >
> > I'd like to thank the authors for the response regarding the contributions on techniques and novelties of the paper (though mainly within the response to Reviewer dgPg), as well as the careful clarifications about the fit-to-venue.
> >
> > I'm not familiar with the monoculture literature, thus I hold a conservative opinion towards the novelty contribution of this paper. I agree with the authors that the straightforward techniques are advantages in the sense of showing the insight. The model contribution seems to firstly extend the literature to the multiple-firm setting, yet weak assumption of homogeneous firms is still not a strong reason for acceptance.
> >
> > Although the supplemented experiments demonstrate that the paper results seem to be correct beyond the homogeneous firms assumption, it seems that for the orientation of this paper, it should be theoretical results rather than experiments that determine the acceptance.
> >
> > Overall, my positive evaluation of this paper remains unchanged.
> >
> > Besides, the meanings of $\beta$ and $\gamma$ are unclear in the attached pdf.

---

> > > ### Author Response · Authors · 2024-08-08
> > >
> > > Thank you for the reply, and we're glad that you found our responses helpful.
> > >
> > > Thanks also for the note about $\beta$ and $\gamma$. These are defined as in lines 743-746 in the paper (in the appendix). $\beta$ controls the level of "global correlation" in applicants' preferences over colleges. $\gamma$ controls the level of "local correlation" (i.e., how much applicants prefer to be close to "nearby" firms).

---

### Official Review · Reviewer_dgPg · 2024-07-15

**Soundness:** 2
**Presentation:** 3
**Contribution:** 1
**Rating:** 3
**Confidence:** 5

**Summary:**

The paper considers a matching model with a continuum of students/applicants and m colleges/firms where the firms have a noisy estimate of the candidates' quality and compares the stable matching outcome in two situations: monoculture (where all firms have the same estimate) vs polyculture (where each firm has an iid estimate). It makes two major assumptions : all firms have the same capacity and candidates' preferences are uniform amongst the m firms. Then the unique stable matching is described by a single cutoff for all firms. The paper states two main results:
- Thm 1: with polyculture, as m grows large, the stable matching approaches an optimal cutoffs mechanism on the true quality; whereas with monoculture it does not
- Thm 2: the probability of first choice match is higher under monoculture
- Thm 3: if candidates can submit variable length preference lists, the students submitting more benefit from polyculture but not from monoculture

**Strengths:**

The topic of the paper is clearly important. From what I see, the paper is not original except in rewording things studied in previous works as monoculture vs polyculture instead of correlation vs independence. Perhaps this can contribute to increasing the volume of literature labeled as studying the effect of monoculture (which is an important concern), but other than that I don't see anything fondamental it brings. The paper is clear. The take-aways are interesting but besides their lack of novelty their significance is diminished a lot by the very strong assumptions made (see below).

**Weaknesses:**

There are two main weaknesses to the paper: (i) lack of novelty in the model, the questions and the flavor of the results and (ii) very strong assumptions that make the results mostly trivial so that I could not identify any strong technical result either. I elaborate on both in the remaining of this box.

- The paper claims (l. 38) that the technical contribution is a matching market model that can be used to analyze monoculture. However, the model used is a particular case of that of [13] (because in [13] they can have multiple demographic groups). Even [13] with a single demographic group is more general because it can handle any level of correlation and not just 0-1 (mono-polyculture). The particular case of latent quality+noise is described in appendix A.4 of [13].

Note that in [13], some results apply only for 2 colleges, but the model applies to any number of colleges.

- The paper makes two very strong assumptions: each firm has the same capacity and candidates' have uniform preferences. Under these, it shows (actually, it just states, because it is trivial) that at the stable matching each firm uses the same cutoff. This makes the rest of the paper technically straightforward, but this is very unrealistic in practice. So, even if one were to see the paper as an extension of some results of [13] for m firms, this would be only under *extremely* simplifying assumptions that make the extension straightforward and of much diminished significance.

Just in contrast, most of the technical difficulty in [13] seems to be in handling the fact that the cutoffs may be different and hence if one increasing, it does not imply that the other does so too. The authors in fact show that the property of diminishing cutoffs is no longer always true for >2 firms. Of course, it holds with all equal cutoffs but as mentioned above, this is too simplifying (and trivial).

- Thm 1 gives an interesting message but it is very straigthforward under the assumptions mentioned above.

- Thm 2 is very similar to [13], the extension to m firms does not seem significant. See my discussion above.

- Thm 3 is very connected to [7]. This is (really) discussed only in the appendix (l. 644-651), but it appears that the additional value of Thm 3 compared to [7] is minimal.

- There are some numerical simulations. Unfortunately, as far as I could see, they do not relax the assumption of same capacity.


Overall, even though the paper is well-written and interesting, I cannot identify any strong or novel result that would get close to the NeurIPS bar. Perhaps if the authors focus on the elements that are novel and try to remove the strong assumptions that make the results straightforward, this would improve the paper's contribution.


[13] Castera et al. EC'22. I used for this review the latest available version from May 2024 https://hal.science/hal-03672270v6, but I checked and the previous version is extremely similar.
[7] Arnosti. MS (2022)

**Questions:**

See weaknesses.

**Limitations:**

I very strongly encourage the authors to be much more upfront about the limitations of their model (discussed at length above). I have not seen them mentioned in abstract or introduction, whereas they are extremely important for the results to hold (in fact, some results do not hold without, see the counter-example at the end of [13] for more than 2 firms).

---

> ### Author Rebuttal · Authors · 2024-08-07
>
> We thank the reviewer for the careful feedback, and we discuss your concerns below.
>
> **Assumptions:** We give detailed comments on our theoretical assumptions in the general response. We reemphasize here that we will incorporate your feedback in being clearer about assumptions in the introduction. We also believe that our results extend beyond the assumptions, and we hope you find the previous experiments (varying correlation of applicant preferences, considering ML experiments) as well as the new experiments (varying firm capacities, as you suggest) convincing of this point.
>
> **Novelty:** We respectfully disagree regarding novelty. We describe three main points of novelty below:
>
> **(1) Modeling true applicant values.** The statement “the model used is a particular case of that of [13]” is inaccurate. This can be seen by observing that our model extends that of Kleinberg and Raghavan, while Castera et al. does not. More specifically, our model departs from Castera et al. (and Arnosti) by following Kleinberg and Raghavan in modeling a ground-truth *true value* of each applicant, which firms estimate. This is essential for many of our results:
> - Theorem 1 reveals that under polyculture, exactly the applicants with the highest true values match.
> - Theorem 2 shows that applicants with high true value face a trade-off between likelihood of matching to top choice (monoculture) and likelihood of matching overall (polyculture).
> - Theorem 3 shows that, under polyculture, applicants with higher true values may fare worse than applicants with lower true values who can apply to more places.
>
> Castera et al. and our work share a common high-level finding with past work on tie-breaking in school choice (correlation improves applicant welfare); however, our respective novel contributions are fundamentally distinct. In particular, Castera et al. certainly identifies interesting and important results that our paper does not touch upon (we discuss these results in lines 110-122 in Section 2); we view our work as studying a complementary set of questions, in studying the effects of monoculture with heterogeneous true applicant values and emergent/strengthening effects as the number of firms grows large
>
> **(2) Bridging literatures.** We also believe that it is a novel contribution to bridge the algorithmic monoculture papers of [Kleinberg and Raghavan (theoretical, firm-side)] and [Bommasani et al. (empirical, applicant-side)] with a matching markets framework. **Prior to our work first being posted online, these literatures did not discuss each other** (the 2022 version of Castera et al. cites Kleinberg and Raghavan only in passing alongside many other algo fairness papers and does not discuss monoculture). As reviewers noted, our work extends and challenges these past ML results. Our ML-based simulations directly show how our framework can be adopted for empirical studies like in Bommasani et al., demonstrating how researchers in ML can utilize the framework.
>
> **(3) M > 2 firms/schools and emergent effects.** All the main results of prior relevant work (including K&R and Castera, who directly state their models in the 2 school setting) hold in a 2 school model. One technical insight of our work is showing how a Azevedo-Leshno continuum model admits a tractable way to study M > 2 schools (Castera also use AL model, but only to study 2 schools). **Crucially, this is not just a technical detail: one important conceptual insight of our work is that considering many firms/schools *fully strengthens* the underlying effect—large polyculture markets behave essentially like noiseless markets (i.e., sort according to true value).** This insight about emergent effects is novel in our work, compared to all the prior literature in both matching markets and machine learning. We also note that, from the proof techniques of Castera and other related work, it is not clear how to derive results for more than 2 schools—doing so seems algebraically intractable. One key insight of our work is to consider maximum-concentrating distributions (which generalizes common Gaussian assumptions), which become extremely tractable with a large number of firms/colleges.
>
> In other words, studying emergent phenomena with many firms is a novel insight of our work, that is distinct from the dimension studied in Castera et al. and other work. To illustrate this, we modify our numerical simulations so that we vary both number of firms and the amount of correlation between firms (which Castera et al. varies). **Indeed, in the bottom figure in the PDF attached in the main response, effects of variation in correlation become stark only when the number of firms is large.**
>
> Finally, we also note that we did not intend to minimize the relationship with Castera et al. Lines 110-120 detail the relationship with Castera, and lines 123-133 summarize some of the above points about emergent phenomena with many firms and the novel aspect of assuming that students have heterogeneous true values. We also have a 3.5 page extended related work section in the Appendix that details the relationship with the burgeoning literature around this area, and our novel contributions.
>
> To summarize, we highly respect the work of Castera et al. and others in this literature, and we believe that our work is complementary, both conceptually and technically, to this line of work. It is inaccurate to say that our model and assumptions nest within past work: while some assumptions (full correlation vs no correlation) are stricter than some past work, other aspects are more general / entirely distinct (especially, considering the true values of applicants and M>>2 firms). In particular, while our model extends Kleinberg and Raghavan, the model in Castera et al., Arnosti, and others do not. We are happy to modify the writing in the main text further to clarify this important point.

---

> > ### Author Response · Authors · 2024-08-08
> >
> > As a brief point of clarification, Castera et al.'s paper does mention latent value, as a potential generating process for differential correlation. However, they do not then analyze applicant outcomes in terms of latent values, as is one of our focuses.

---

> > ### Comment · Reviewer_dgPg · 2024-08-08
> >
> > I thank the authors for their response. This does not change my view of the paper for a number of reasons. First, it is not true that [13] does not model true value (which they call latent value), as the added comment of the authors seems to acknowledge. It is correct, as the authors state, that [13] does not focus on this in the results, but still, this (to me) considerably reduces the novelty of the model. Also, regarding m colleges (instead of 2): the model of [13] is stated with 2 colleges, however they mention in the discussion that the model trivially extends to m colleges; so I was not able to understand the sentence stated in the rebuttal, "One technical insight of our work is showing how a Azevedo-Leshno continuum model admits a tractable way to study M > 2 schools".
> >
> > Regarding the results, some results are certainly different in their statement than past work, however as I have mentioned in my review, I find the results extremely underwhelming given the extremely strong assumptions. I was not convinced by the argument that strong assumptions are a feature rather than a limitation. Also, I appreciate that the authors made extra simulations with random capacities, but I find the outcome quite unreliable. Indeed, I do not think that it is reasonable to draw any conclusion that "the theoretical results continue to hold" with such simulations. First, random settings are often not the pathological ones that challenge the theoretical results. Second, they are also not happening in practice. Finally, note that [13] has a counter-example in the discussion showing that with 4 colleges the property of diminishing cutoffs may not hold. This counter-example has capacities (0.05, 0.05, 0.2, 0.5) if I read well Fig. 5, which does not seem to be random. This seems to indicate that pathological things may happen and showing numerical simulation with random capacities does not rule them out.
> >
> > All in all, I agree that the paper has some results that were not explicitly stated that way in past literature, but I still find that the contribution is insufficient in terms of novelty and technical significance.

---

> > > ### Author Response · Authors · 2024-08-09
> > >
> > > We respond to each point below. Fundamentally, we believe that Castera et al. (a nice paper!!) studies a different question than us, our results/insights/techniques are conceptually new, and that another paper stating that their model can extend to ours does not equate to **findings** that affect novelty.
> > >
> > > **“First, it is not true that [13] does not model true value (which they call latent value), as the added comment of the authors seems to acknowledge.”**
> > >
> > > Again, we disagree with this comment. The model in [13] is only stated in terms of estimated preferences (which can be *motivated* by a latent value model). Consequently, and most importantly, no results imply anything about students in terms of their latent value, as all our results do.
> > >
> > > **“Also, regarding m colleges (instead of 2): the model of [13] is stated with 2 colleges, however they mention in the discussion that the model trivially extends to m colleges; so I was not able to understand the sentence stated in the rebuttal, ‘One technical insight of our work is showing how a Azevedo-Leshno continuum model admits a tractable way to study M > 2 schools’.”**
> > >
> > > While it is of course true that the model in [13] could be stated with many colleges, we believe the relevant factor is if the **results** are extended.
> > >
> > > Castera et al. does not prove many-firm results; we show that some effects *only emerge* when considering many schools (for example, comparing to results in Kleinberg and Raghavan). Moreover, increasing the number of schools *increases theoretical tractability,* especially when combined with studying maximum-concentrating noise (e.g., Gaussian, bounded).
> > >
> > > **Concerns about experimental setup: “First, random settings are often not the pathological ones that challenge the theoretical results. Second, they are also not happening in practice.”**
> > >
> > > We believe that (1) our numerical experiments capture the important/realistic types of variations in real markets, and (2) our ML-based experiments (extending those of Bommasani et al.) are more realistic than existing theory/simulations and speak to the particular community of interest.
> > >
> > > The focus of our experiments is not to identify potential “pathological” counterexamples, but rather to test our predictions in a range of settings that we believe mimic real markets. Our simulation setup focuses on two types of correlation widely noted in the literature: correlation arising from shared vertical preferences, as well as from horizontal preferences (preferences that depend on “proximity”). These are controlled by $\beta$ and $\gamma$ parameters (see lines 740-746), which we vary widely. We also simulate firm preferences generated using ML models—testing our predictions in a more realistic and relevant decision-making setting.
> > >
> > > More broadly, we think a key role of a model is to convey intuition and provide useful predictions. For example, even though Castera et al. note that “most of our results do not extend to more than two colleges” (page 5), we think that their two-college results are useful to understand broader phenomena (e.g., increasing correlation in one group can help all groups). In fact, Fig. 5 in their paper, even if it does not *exactly* match their theory, seems to suggest that the theory a very good approximation in this “pathological” example.

---

### Official Review · Reviewer_V2nV · 2024-08-05

**Soundness:** 3
**Presentation:** 3
**Contribution:** 3
**Rating:** 6
**Confidence:** 4

**Summary:**

This paper examines the effects of algorithmic monoculture in a large two-sided matching market, in which participants on both sides compete with each other and outcomes are determined by preferences on both sides.
It proposes a matching markets model to study monoculture and produces both expected and surprising results.
While under monoculture, all firms use a single shared estimate of an applicant's value, under polyculture, they obtain separate, independently drawn estimates.
- One expected result is that polyculture benefits firm welfare.
- More surprisingly, another result shows that applicants are better off under monoculture, yet risk-averse well-qualified applicants would rather prefer the security afforded by polyculture. In other words, and expectedly in this regard, monoculture presents a risk of systemic exclusion to certain more qualified applicants.
- A third result is that polyculture benefits applicants who submit more applications, thus allows differences in the number of applications submitted to harm firm welfare.

**Strengths:**

S1. Connects literature on monoculture and matching markets.
S2. Produce novel results.
S3. Computational experiments training on real data.

**Weaknesses:**

W1. The results in Figure 5 seem to contradict the strong claims made about polyculture outperforming monoculture; the related claims need to be qualified.
W2. The concept of "positive label", or binary 0-1 outcome, is not defined.
W3. In the experiment, applicants have uniformly random preferences, which do not yield competition as suggested.

**Questions:**

What happens when applicants have competitive preferences?

**Limitations:**

Yes.

---

> ### Author Rebuttal · Authors · 2024-08-07
>
> Thanks for the helpful comments. We’re glad to see that you found the results interesting, and at times surprising. We address your comments, mostly about the ML experiments, below:
>
> “The results in Figure 5 seem to contradict the strong claims made about polyculture outperforming monoculture; the related claims need to be qualified.”
> - Thanks, we agree that the results in Figure 5 (California data) are less strong than those in Figure 3 (Texas data), and that we should qualify the claims more in the main text. The current paper briefly notes that results are weaker, but we will add exact description of results into the main text. Specifically, we will move line 709-714 in the appendix into the main text. We will also more carefully note that the weaker results could be due to less adherence to our “maximum-concentrating noise” assumption of Theorem 1, and that the accuracy of this assumption needs to be further studied in real-world settings.
>
> “The concept of ‘positive label’, or binary 0-1 outcome, is not defined.”
> - Thanks, we will add more description. First, we will note that the 0-1 label in the ACSIncome dataset is whether or not an applicant exceeds a certain income. We will also note that this itself is not a realistic task for make hiring decisions, but is a standard ML prediction task similar to those used in hiring. In particular, the ML experiments mimic the setup of Bommasani et al., except with many firms in a market framework.
>
> “In the experiment, applicants have uniformly random preferences, which do not yield competition as suggested.” “What happens when applicants have correlated preferences?”
> - We further tested the ML experiments in a setting where applicants have shared preferences over firms, regenerating Figures 3 and 5. These can be seen in our PDF. The results continue to hold, almost identically. We will make note of this robustness check in the paper.
> - More generally, we substantially vary the amount and type of variation in the numerical simulations (as noted in Section 5, and fully detailed in Appendix C). We find that each of our (directional) theoretical predictions continue to hold over all parameterizations.

---

> > ### Comment · Reviewer_V2nV · 2024-08-14
> >
> > Your reply should mention whether there is something to be seen in the current pdf, and if so, where that is to be seen.
> > The statement "these can be seen in our PDF" is not helpful.

---

> > > ### Author Response · Authors · 2024-08-14
> > > **PDF details**
> > >
> > > We apologize. The PDF reference regarding applicants having shared preferences over firms refers to the middle pair of plots, labeled "Texas, Correlated Applicant Prefs" and "California, Correlated Applicant Prefs". Thank you for considering our response!

---

> > > > ### Comment · Reviewer_V2nV · 2024-08-14
> > > >
> > > > The respose remains unhelpful; such plots do not exist in the provided pdf.

---

> > > > > ### Author Response · Authors · 2024-08-14
> > > > > **Plot/pdf location**
> > > > >
> > > > > Hi, I think there might be some confusion. The pdf we are referring to is the one page pdf attached to the author rebuttal to all reviewers at the top. In the middle of the pdf there are 2 sets of plots next to large text on the left that says "Texas, Correlated Applicant Prefs" and "California, Correlated Applicant Prefs", respectively. These are the plots where the x axis says "subsets of models" and y axes say "accuracy" or "average rank of match". There are many red and blue dots in these plots. These replicate Figures 3 and 5 of the initial submission, showing generally that under monoculture applicants rank to their more preferred matches, even when their preferences are correlated.

---

### Author Rebuttal · Authors · 2024-08-07

We thank all the reviewers for the thoughtful comments.

We’re glad to see that reviewers found the paper to be clear and well-written (R2, R3, R4), and the results to be surprising, countering past literature and expectations about algorithmic monoculture (R1, R3, R4). All reviewers noted how the model bridges the ML literature with the matching markets literature.

The most common concerns were about the strength of theoretical assumptions (particularly noted by R2, and by others to varying degrees); R2 relatedly noted that this led to simple and straightforward proofs.

In this general response, we discuss these concerns about theoretical assumptions (and corresponding simple proofs). We defer more reviewer-specific questions and concerns to individual responses, including concerns about novelty (R2) and fit-to-venue (R3).

**Discussion on theoretical assumptions:** We agree that our theoretical model makes a strong assumption of homogeneous firms (equal capacities, and that students have uniformly random preferences). At a high level, we believe that our experiments convincingly demonstrate that the conceptual insights extend beyond these assumptions.

Both R2 and R4 note that we should make this assumption clearer earlier in the text. We fully agree, and will add the following after line 46 in the introduction: “Our theoretical results are shown in a setting with a symmetric setup with a large number of homogeneous firms. This allows for tractability, as well as clear intuition. We demonstrate empirically that our theoretical results hold in markets with heterogeneous firms, as well as when firms use ML models to rank applicants.”

To address limitations in our theoretical model, we test our theoretical predictions in computational experiments. In our paper, we included both (1) simulations of markets with correlated student preferences, following a simulation setup of Ashlagi et al., and (2) simulations of markets with ML models, bringing the setup in Bommasani et al. to a market context. Each of these experiments verify the predictions of our model.

R2 noted that our theoretical results on polyculture only consider fully independent noise. Our ML experiments represent a case in which firms using different algorithms have some but not perfect correlation (since the ML models share data and overlap in features).

There were still some settings that our experiments did not cover, and which reviewers noted. To alleviate these concerns, we include a PDF of additional experiments addressing these specific concerns:
- R1 noted that our ML experiments only assumed uniformly random preferences (unlike our pure simulation setting). We include the corresponding figures when applicants have heavily-correlated preferences (beta=10 in our simulation setup), showing that the same results hold.
- R2 noted that our numerical experiments did not vary capacity. We rerun each numerical experiment when firms have random differing capacities (still also varying correlation in applicant preferences), showing that our results continue to hold. In the paper, we will remark that these further experiments align with our theoretical results. We note that varying capacity has similar effects as non-uniform preferences (which our simulation setting already includes), as they induce heterogeneous cutoffs.

**Straightforwardness.** While this theoretical setup makes our proofs relatively straightforward (as R2 notes), we believe this to be a feature, and that our work makes a significant technical contribution in two ways. In particular, our model *generalizes* aspects of models in past work that make the model *more tractable:* we consider many firms rather than two firms (as considered by Kleinberg and Raghavan, and Castera et al.). We also consider an interpretable-but-general class of noise distributions (that includes bounded and Gaussian distributions common in other work). The key technical insight is that by considering many firms, we can leverage the well-behaved tail of max-concentrating distributions to tractably (and cleanly) analyze the resulting stable matching. The simple-seeming nature of our proofs is a consequence of these technical insights, contra the (substantial) algebra often needed in prior work.

We also want to note the attached pdf, in which we run new computational experiments to supplement our responses, including around assumptions and to make clearer the relationship with prior work.

---

### Decision · Program_Chairs · 2024-09-25

**Decision:**

Accept (poster)

**Comment:**

There were three positive reviews (5, 6, 7) and one quite negative review (3) on this paper. The paper is exceptionally well-written and nicely presented, and although the model is very simplistic, leading to straightforward proofs (not necessarily a bad thing, as simple proofs more clearly show the intuition behind them), the theoretical results are somewhat surprising. Furthermore, the authors take great pains to do extensive simulations to show that their results extend past their simple theoretical model.

The negative review (dgPg) focuses on lack of novelty in the model---in particular, they claim that it is a specific case of [Castera et al., EC 2022]---and the simplicity of the theoretical results. For the simplicity concern, as stated above, I agree with LFB2 that simple proofs are sometimes desirable. As to the lack of novelty, the authors effectively rebutted dgPg's concerns that their model is a straightforward extension of previous work (it is not), and the results they were able to prove are significantly different from those of [Castera et al., 2022].

Overall, I feel like this paper's strengths (extremely well-written, empirical results that go beyond their theoretical model) outweigh its weaknesses (simple model, straightforward results), especially once the authors clarified its novelty compared to previous work.